# ReMA: Learning to Meta-think for LLMs with Multi-agent Reinforcement Learning

**Ziyu Wan**[1,2]*[†], **Yunxiang Li**[3]*, **Xiaoyu Wen**[1,2]*, **Yan Song**[4], **Hanjing Wang**[1], **Linyi Yang**[4], **Mark Schmidt**[3,5], **Jun Wang**[4], **Weinan Zhang**[1], **Shuyue Hu**[2][‡], **Ying Wen**[1][‡]

[1] Shanghai Jiao Tong University
[2] Shanghai Artificial Intelligence Laboratory
[3] University of British Columbia
[4] University College London
[5] Canada CIFAR AI Chair (Amii)

## Abstract

Recent research on Reasoning of Large Language Models (LLMs) has sought to further enhance their performance by integrating meta-thinking—enabling models to monitor, evaluate, and control their reasoning processes for more adaptive and effective problem-solving. However, current single-agent work lacks a specialized design for acquiring meta-thinking, resulting in low efficacy. To address this challenge, we introduce **Re**inforced **M**eta-thinking **A**gents (ReMA), a novel framework that leverages Multi-Agent Reinforcement Learning (MARL) to elicit meta-thinking behaviors, encouraging LLMs to *think about thinking*. ReMA decouples the reasoning process into two hierarchical agents: a high-level meta-thinking agent responsible for generating strategic oversight and plans, and a low-level reasoning agent for detailed executions. Through iterative reinforcement learning with aligned objectives, these agents explore and learn collaboration, leading to improved generalization and robustness. Empirical results from single-turn experiments demonstrate that ReMA outperforms single-agent RL baselines on complex reasoning tasks, including competitive-level mathematical benchmarks and LLM-as-a-Judge benchmarks. Additionally, we further extend ReMA to multiturn interaction settings, leveraging turn-level ratio and parameter sharing to improve efficiency. Comprehensive ablation studies further illustrate the evolving dynamics of each distinct agent, providing valuable insights into how the meta-thinking reasoning process enhances the reasoning capabilities of LLMs. Our code can be found in `https://github.com/ziyuwan/ReMA-public`

## 1 Introduction

Large language models (LLMs) have demonstrated remarkable capabilities in knowledge understanding and complex reasoning tasks [Chowdhery et al., 2023, Achiam et al., 2023, Anil et al., 2023, Dubey et al., 2024]. The paradigm in developing LLM-based reasoning models is shifting from scaling training-time computation towards scaling test-time computation [Snell et al., 2024]. Recent advancements, such as OpenAI-o1 [OpenAI, 2024], Deepseek R1 [DeepSeek-AI et al., 2025], and Gemini 2.0 Flash Thinking [DeepMind, 2025], have demonstrated that allowing LLMs to think before generating answers can significantly enhance performance and lead to the emergence of

---

*Equal contribution.
†Work done during internship at Shanghai Artificial Intelligence Laboratory
‡Corresponding Author

39th Conference on Neural Information Processing Systems (NeurIPS 2025).

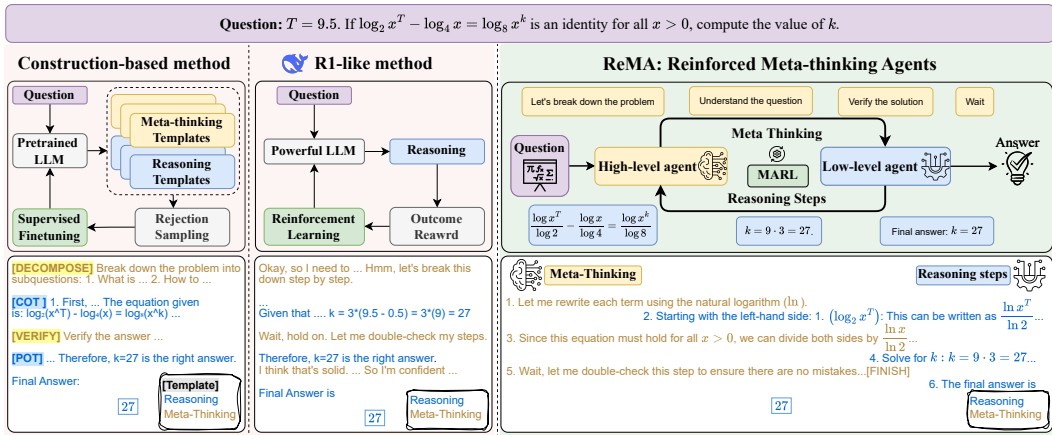

Figure 1: Left: A construction-based method that fine-tunes LLMs using rejection sampling, searching among combinations of pre-defined templates. Middle: R1-like method learns to mix meta-thinking and detailed reasoning steps during training. Right: Our method ReMA separates the meta-thinking and reasoning steps in a multi-agent system and updated by reinforcement learning.

human-like reasoning patterns. These patterns like **"Wait, hold on."** or **"Let's break this down."**, indicate that LLMs can develop a form of *meta-thinking* abilities that can generalize well to out-of-distribution (OOD) tasks [Xiang et al., 2025]. Meta-thinking, also known as metacognitive skills [Flavell, 1979], is an ability traditionally considered uniquely human [Didolkar et al., 2024].

To cultivate meta-thinking patterns from LLMs themselves, recent construction-based supervised approaches leverage supervised finetuning on structured reasoning trajectories. Specifically, these methods sampling reasoning trajectories from predefined meta-thinking templates and then use supervised finetuning (SFT) or direct preference optimization (DPO) [Rafailov et al., 2023] to teach LLMs imitate these patterns [Qi et al., 2024, Yue et al., 2024, Xi et al., 2024, Yang et al., 2025, Muennighoff et al., 2025, Ye et al., 2025c]. However, such methods lack sufficient flexibility for LLMs to explore suitable meta-thinking patterns. Thus, they often fail to generalize to out-of-distribution (OOD) problems, leading to unstable performance on unseen data [Kirk et al., Chu et al., 2025]. Besides construction-based methods, R1-like single-agent reinforcement learning (SARL) has also been adopted for meta-thinking in reasoning [DeepSeek-AI et al., 2025, Xie et al., 2025]. However, these SARL attempts typically rely on strong foundational models for easier exploration or extensive task-specific fine-tuning for stable training [Xu et al., 2025, Gandhi et al., 2025]. Furthermore, SARL needs to learn meta-thinking and reasoning within a single forward pass, seeking to capture complex reasoning structures purely in an autoregressive manner [Xie et al., 2025]. This can potentially lead to issues such as inefficient exploration as well as reduced readability and early convergence to local optima [DeepSeek-AI et al., 2025, Xiang et al., 2025].

To address these limitations, we introduce **Re**inforced **M**eta-thinking **A**gents (ReMA), a novel framework that leverages multi-agent reinforcement learning (MARL) to encourage LLMs to *think about thinking*. Our approach employs a multi-agent system (MAS) composed of a high-level **meta-thinking** agent, responsible for strategic oversight and instruction generation, and a low-level **reasoning** agent tasked with detailed executing processes based on provided guidance. We compare the inference process among the construction-based method, R1-like method, and ReMA in Fig. 1. Since MAS distributes the exploration space of SARL into multiple agents, it enables each agent to explore more structurally and efficiently during training. Then we apply reinforcement learning on each agent with aligned reward functions. In this way, ReMA effectively balances the trade-off between generalization capability and exploration efficiency. As a result, they can learn to play the best of their role (either to meta-think or to follow instructions), at the present of the other agent.

To our knowledge, we are the first to formally define and optimize a multi-agent meta-thinking reasoning process (MAMRP) through multi-agent reinforcement learning. Our extensive experiments span both math reasoning and LLM-as-a-Judge tasks, where ReMA consistently achieves the highest average performance across three backbone pretrained models. We further extend ReMA to multi-turn interaction settings on math reasoning tasks, implementing turn-level ratio to optimize

trajectory returns and stabilize training. Through comprehensive ablation studies, we illustrate the evolving dynamics between agents, revealing unexpected interaction patterns such as role reversals under different reward settings. These findings provide valuable insights into how meta-thinking processes enhance the reasoning capabilities of LLMs.

## 2 Preliminaries

In this section, we outline the formulation of the vanilla reasoning process (Sec. 2.1) and the representative training methods (Sec. 2.2) along with the notation used throughout the paper.

### 2.1 Vanilla Reasoning Process (VRP)

The probability of generating a response $\mathbf{y}$ equals the product of its stepwise probabilities. Given a model $\pi_\theta$ and a prompt $\mathbf{x} = (x_1, \ldots, x_N)$, the **vanilla reasoning process** (VRP) autoregressively produces a response $\mathbf{y} = (y_1, \ldots, y_L)$ with

$$\pi_\theta(\mathbf{y}|\mathbf{x}) = \prod_{l=1}^{L} \pi_\theta(y_l|x_1, x_2, \ldots x_N, y_1, \ldots, y_{l-1}) = \prod_{l=1}^{L} \pi_\theta(\mathbf{y}_l|\mathbf{x}, \mathbf{y}_{<l})$$

The response usually contains intermediate reasoning steps before arriving at the final answer, this process is also known as **chain-of-thought** (CoT) [Wei et al., 2022], which can be represented as:

$$\mathbf{x} \xrightarrow{\text{reasoning steps}} \mathbf{y} \sim \mathbf{a}, \tag{1}$$

where $\mathbf{a}$ is the extracted final answer, which is included in the answer $\mathbf{y}$.

### 2.2 Training VRP via Reinforcement Learning

RL frames VRP decoding process as a *deterministic*, token-level Markov Decision process (MDP) [Wang et al., 2024a]. Its objective is

$$\mathcal{J}(\theta) = \mathbb{E}_{(\mathbf{x},\mathbf{y}^*)\sim\mathcal{D},\mathbf{y}\sim\pi_\theta} \left[ R(\mathbf{y}, \mathbf{y}^*) \right].$$

where $R(\cdot, \cdot)$ represents a reward function comparing generated answer $\mathbf{y}$ with the golden answer $\mathbf{y}^*$ for any question $\mathbf{x}$ sampled from dataset $\mathcal{D}$.

To compute the gradient $\nabla_\theta \mathcal{J}(\theta)$, computationally efficient algorithms GRPO [Shao et al., 2024] and REINFORCE++ [Hu, 2025] are widely adopted. Take GRPO as an example, given a question-answer pair $\mathbf{x}, \mathbf{y}^*$ and a group of $G$ generated responses $\mathbf{y}_i$, denote $\mathbf{y}_{i,j}$ as the $j$-th token of the $i$-th response, it optimizes the following token-level objective:

$$\mathcal{J}(\theta) = \mathbb{E}_{(\mathbf{x},\mathbf{y}^*)\sim\mathcal{D},\ \{\mathbf{y}_i\}_{i=1}^{G}\sim\pi_{\theta_{\text{old}}}(\cdot|\mathbf{x})}$$
$$\left[ \frac{1}{G}\sum_{i=1}^{G}\frac{1}{|\mathbf{y}_i|}\sum_{j=1}^{|\mathbf{y}_i|}\Big(\min\big(r_{i,j}(\theta)\,\hat{A}_{i,j},\,\text{clip}\big(r_{i,j}(\theta),\,1-\epsilon,\,1+\epsilon\big)\,\hat{A}_{i,j}\big) - \beta\,D_{\text{KL}}\big(\pi_\theta\|\pi_{\text{ref}}\big)\Big) \right], \tag{2}$$

where the token-level ratio $r_{i,j}(\theta)$ and the group-normalized advantage $\hat{A}_{i,j}$ are defined as:

$$r_{i,j}(\theta) = \frac{\pi_\theta\big(\mathbf{y}_{i,j} \mid \mathbf{x},\,\mathbf{y}_{i,<j}\big)}{\pi_{\theta_{\text{old}}}\big(\mathbf{y}_{i,j} \mid \mathbf{x},\,\mathbf{y}_{i,<j}\big)}, \hat{A}_{i,j} = \frac{R_i - \text{mean}(\{R_i\}_{i=1}^{G})}{\text{std}(\{R_i\}_{i=1}^{G})}.$$

However, RL on base LLMs that haven't been well-aligned may suffer from issues like poor readability and language mixing, preventing researchers from verifying, understanding, and further developing their LLMs. And huge searching space makes efficient learning of meta-thinking daunting.

## 3 Method

In this section, we present **Re**inforced **M**eta-thinking **A**gents (ReMA), a RL method integrating meta-thinking into the reasoning process of LLM under multi-agent settings (Sec. 3.1), then describe the learning process enabled by MARL of single- and multi-turn LLM setting (Secs. 3.2.1 and 3.2.2).

## 3.1 Deploying Meta-Thinking Reasoning Process for LLMs

Beyond VRP (Sec. 2.1), recent studies [Muennighoff et al., 2025, Ye et al., 2025c] have shown that integrating meta-thinking behaviors in reasoning process can largely improve the accuracy of the final answers. By integrating Meta-thinking, ReMA decomposes problem solving into two sequential phases: a **meta-thinking** phase that plans, monitors, or revises strategy, followed by a **reasoning** phase that produces the detailed solution. We analyse Meta-thinking Reasoning Process along two orthogonal axes—*single- vs. multi-agent* and *single- vs. multi-turn*.

In a single-agent setting, such a process calls LLM once and generates meta-thinking and the following reasoning autoregressively. We formulate the **meta-thinking reasoning process** (MRP) below:

$$\mathbf{y} \sim \pi_\theta(\mathbf{y} \mid \mathbf{x}, \mathbf{m}) \cdot \pi_\theta(\mathbf{m} \mid \mathbf{x}), \tag{3}$$

where $\mathbf{m}$ and $\mathbf{y}$ are the output of meta-thinking and reasoning respectively. We present the procedure as shown below:

$$\mathbf{x} \xrightarrow{\text{meta-thinking}} \mathbf{m} \xrightarrow{\text{reasoning steps}} \mathbf{y} \sim \mathbf{a}. \tag{4}$$

Exploring MRP reasoning through a single-agent approach is often inefficient, as it requires the language model to simultaneously master both meta-thinking and detailed problem-solving *within one call*. Prior research has demonstrated that activating different model capabilities through specialized agents significantly improves MRP exploration efficiency. To leverage this insight, we decouple meta-thinking and reasoning into two separate LLM agents: a high-level agent dedicated to generating meta-thinking, and a low-level agent focused on executing reasoning steps.

During a conversation, the high-level and low-level agents (*i.e.*, $\pi_h$ and $\pi_l$) act in an interleaving manner. The high-level agent generates and summarizes meta-thoughts from the prompt and interaction history, while the low-level agent executes detailed problem-solving under those instructions. We formulate the **multi-agent meta-thinking reasoning process** (MAMRP) as follows:

$$\mathbf{y} \sim \pi_l(\mathbf{y} \mid \mathbf{x}, \mathbf{m}) \, \pi_h(\mathbf{m} \mid \mathbf{x}). \tag{5}$$

While the single-turn MAMRP offers a straightforward approach, it lacks the ability to perform immediate and fine-grained cognitive switching during the reasoning process, which limits its effectiveness on complex, long-horizon planning tasks. Therefore, we extend Eq. (5) and formulate the multi-turn MAMRP as follows:

$$\mathbf{y}_T \sim \prod_{t=1}^{T} \pi_l(\mathbf{y}_t \mid \mathbf{x}, \{\mathbf{m}, \mathbf{y}\}_{<t}, \mathbf{m}_t) \, \pi_h(\mathbf{m}_t \mid \mathbf{x}, \{\mathbf{m}, \mathbf{y}\}_{<t}) \tag{6}$$

where $T$ is the number of turns. Similarly, we present the process with a directed graph:

$$\mathbf{x} \xrightarrow[\pi_h]{\text{meta-thinking}} \mathbf{m}_1 \xrightarrow[\pi_l]{\text{reasoning}} \mathbf{y}_1 \xrightarrow[\pi_h]{\text{meta-thinking}} \mathbf{m}_2 \xrightarrow[\pi_l]{\text{reasoning}} \mathbf{y}_2 \xrightarrow[\pi_h]{\text{meta-thinking}} ... \xrightarrow[\pi_l]{\text{reasoning}} \mathbf{y}_T \sim \mathbf{a}. \tag{7}$$

As a complex reasoning system, MAMRP provides various optimization opportunities in scaling inference-time computation. We leave further discussion of these aspects in Appendix C.1.

## 3.2 Training MAMRP: A Multi-Agent RL Method

Multi-agent RL, unlike single-agent RL in a *deterministic* MDP, must contend with *stochastic*, *non-stationary* dynamics and rewards, making optimization more challenging. We start by considering **an easier case**, the optimization of single-turn MAMRP.

### 3.2.1 Optimizing Single-turn MAMRP

To train the system from Sec. 3.1, we embed it as a Markov Game between the two agents. Suppose the two LLM agents are parameterized by $\theta_h$ and $\theta_l$, respectively. Define a joint hierarchical policy over sequential decisions $\mathbf{m}$ and $\mathbf{y}$:

$$\mathbf{y} \sim \pi_{(\theta_h, \theta_l)}(\mathbf{y} \mid \mathbf{x}) := \pi_{\theta_l}(\mathbf{y} \mid \mathbf{x}, \mathbf{m}) \cdot \pi_{\theta_h}(\mathbf{m} \mid \mathbf{x}), \tag{8}$$

Let $R(\mathbf{y}, \mathbf{y}^*)$ denote the final reward serves as the objective function $\mathcal{J}(\theta_h, \theta_l)$ for the joint policy:

$$\mathcal{J}(\theta_h, \theta_l) = \mathbb{E}_{\mathbf{x}, \mathbf{y}^*} \mathbb{E}_{\mathbf{y} \sim \pi_{(\theta_h, \theta_l)}} R(\mathbf{y}, \mathbf{y}*). \tag{9}$$

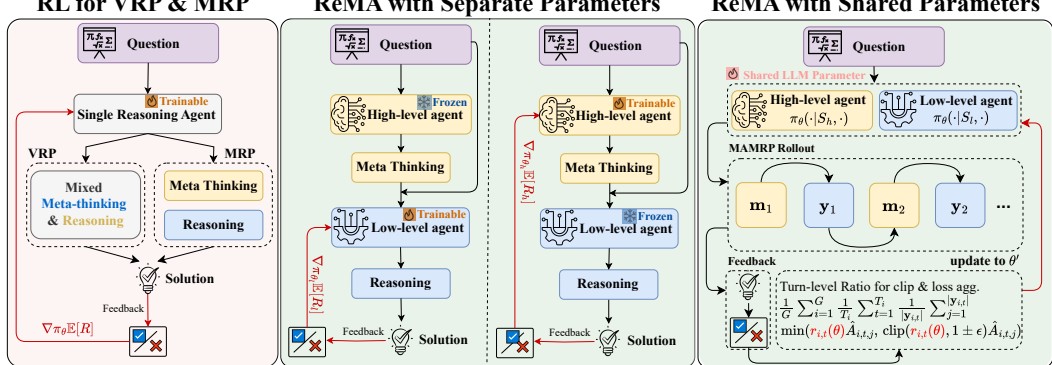

Figure 2: Comparison of training pipelines. **Left:** RL training of VRP and MRP, where a single LM agent is updated either with mixed (VRP) or explicit (MRP) meta-thinking. **Middle:** ReMA with separate parameters for the high-level (meta-thinking) and low-level (reasoning) agents; training alternates between freezing one agent and updating the other. **Right:** ReMA with shared parameters and multi-turn interactions: both agents share the same parameters and are distinguished by their system prompts. Training employs a turn-level ratio for stable multi-turn reinforcement learning and efficient updates, ensuring each turn's contribution is controlled to prevent instability.

During optimization procedure, the high-level policy $\pi_{\theta_h}$ and low-level policy $\pi_{\theta_l}$ aim to maximize their respective rewards independently. The optimization goals for agents are:

$$\theta_h^* = \arg\max_{\theta_h} \mathbb{E}_{(\mathbf{x},\mathbf{y}^*)\sim\mathcal{D},\mathbf{m}\sim\pi_{\theta_h},\mathbf{y}\sim\pi_{\theta_l^*}} \left[ R_h(\mathbf{m},\mathbf{y},\mathbf{y}^*) \right], \tag{10}$$

$$\theta_l^*(\theta_h) = \arg\max_{\theta_l} \mathbb{E}_{(\mathbf{x},\mathbf{y}^*)\sim\mathcal{D},\mathbf{m}\sim\pi_{\theta_h},\mathbf{y}\sim\pi_{\theta_l}} \left[ R_l(\mathbf{m},\mathbf{y},\mathbf{y}^*) \right], \tag{11}$$

where $R_h$ and $R_l$ are policies' individual reward functions, including $R$ and regularization according to tasks and models, e.g., different format rewards (refer to Appendix C.2 for details). The detailed algorithm is in the Algorithm 1. We illustrate the MAMRP inference procedure and the proposed training method in Fig. 2. We also provide an analysis of different loss functions in Appendix C.5.

### 3.2.2 Scaling up to Multi-turn MAMRP

To scale up to multi-turn MAMRP, we can still adopt the iterative training strategy in Sec. 3.2.1. However, we make some changes to improve the efficiency of rollout and training.

First, we implement a parameter-sharing strategy where both high-level and low-level agents utilize identical model weights $\theta$, distinguished only by role-specific system prompts $S_h$ and $S_l$. Formally, we define $\pi_h = \pi_\theta(\cdot|S_h,\cdot)$ and $\pi_l = \pi_\theta(\cdot|S_l,\cdot)$, sharing the same underlying parameters rather than maintaining separate model instances. This approach eliminates the need for frequent model swapping on GPU during rollout, avoiding inefficient wait times, while enabling larger batch sizes during training to simultaneously optimize policies for both meta-thinking and reasoning roles.

Second, we propose a multi-turn GRPO with **turn-level ratio** to address the challenges of multi-turn MAMRP. The trajectory-level averaged objective with **turn-level ratio** of $\pi_l$ is defined as (The objective of $\pi_h$ is the similar but with different system prompt):

$$\mathcal{J}(\theta) = \mathbb{E}_{(\mathbf{x},\mathbf{y}^*)\sim\mathcal{D},\{(\mathbf{m}_i,\mathbf{y}_i)\}_{i=1}^G\sim\pi_{\theta_{\text{old}}}(\cdot|\mathbf{x})}$$

$$\left[ \frac{1}{G}\sum_{i=1}^G \frac{1}{T_i}\sum_{t=1}^{T_i} \frac{1}{|\mathbf{y}_{i,t}|}\sum_{j=1}^{|\mathbf{y}_{i,t}|} \left( \min(r_{i,t}(\theta)\hat{A}_{i,t,j}, \text{clip}(r_{i,t}(\theta),1-\epsilon,1+\epsilon)\hat{A}_{i,t,j}) - \beta D_{\text{KL}}(\pi_\theta\|\pi_{\text{ref}}) \right) \right] \tag{12}$$

where $\mathbf{y}_{i,t,j}$ is the $j$-th token at turn $t$ of the reasoning agent of the $i$-th trajectory. And the turn-level ratio for clipping is defined as:

$$r_{i,t}(\theta) = \exp\Big(\frac{1}{|\mathbf{y}_{i,t}|}\sum_{j=1}^{|\mathbf{y}_{i,t}|}\log r_{i,t,j}(\theta)\Big) \tag{13}$$

$$= \exp\Big(\frac{1}{|\mathbf{y}_{i,t}|}\sum_{j=1}^{|\mathbf{y}_{i,t}|}\log \frac{\pi_\theta\big(\mathbf{y}_{i,t,j} \mid \mathbf{x}, \{\mathbf{m}_{i,\cdot}, \mathbf{y}_{i,\cdot}\}_{<t}, \mathbf{m}_{i,t}, \mathbf{y}_{i,t,<j}\big)}{\pi_{\theta_{\mathrm{old}}}\big(\mathbf{y}_{i,t,j} \mid \mathbf{x}, \{\mathbf{m}_{i,\cdot}, \mathbf{y}_{i,\cdot}\}_{<t}, \mathbf{m}_{i,t}, \mathbf{y}_{i,t,<j}\big)}\Big). \tag{14}$$

The introduction of the turn-level ratio serves two key purposes. First, using a token-level ratio (Eq. (2)) in the objective introduces bias for multi-turn training, as it averages over all tokens in a trajectory. This means that tokens within longer turns (those containing more tokens) can disproportionately influence the overall loss, and averaging at the token level may encourage excessively long single-turn responses. Second, clipping each token independently risks instability during training.

In contrast, the turn-level ratio aligns more closely with the underlying MDP formulation by treating all tokens within a turn as a single action and applying clipping at the turn level. Intuitively, this approach stabilizes training by preventing the LLM from making unstable updates that could result in extreme outputs, such as overly long repetitions or incoherent text. We conduct experimental verification in subsequent empirical results (Sec. 4.3).

## 4 Experiments

To evaluate the effectiveness and efficiency of ReMA, we conduct experiments on challenging benchmarks for two types of tasks: mathematical reasoning and LLM-as-a-Judge with three different LLMs. Then, we investigate the models' performance in both single- & multi-turn settings. Finally, we provide ablation studies and qualitative analyses of our method.

### 4.1 Experiment Settings

We first analyze the single-turn case of ReMA, *i.e.*, $T = 1$. The high-level agent generates a complete meta-thinking trace in one shot, and the low-level agent follows the instructions and outputs the final results. Single-turn ReMA reduces stochasticity and training cost while our experiments show that it still provides meaningful performance gains.

**Benchmarks** We conduct experiments on two types of tasks: mathematical reasoning and LLM-as-a-Judge. For mathematical reasoning experiments, we train models on 7.5k training samples in MATH [Hendrycks et al., 2021] and use MATH500 [Lightman et al., 2023] as the in-distribution test dataset. Additionally, we test the optimized models on out-of-distribution datasets: GSM8K [Cobbe et al., 2021], AIME24[4], AMC23[5], GaoKao2023En [Zhang et al., 2023], Minerva Math [Lewkowycz et al., 2022], and Olympiad Bench [He et al., 2024].

For LLM-as-a-Judge benchmarks, we train models on RewardBench [Lambert et al., 2024]. Specifically, we convert the original data into a pair-ranking format and split it into a training set of 5k items and a test set of 970 items, denoted as RewardBench970. The models are also tested on JudgeBench [Tan et al., 2024] to assess out-of-distribution performance. We refer to Appendix D.1.2 for detailed comparisons and results.

**Baselines, Models, Training Settings** We compare pass@1 performance across the following methods: (1) **VRP** (CoT, step-by-step prompting, Sec. 3.1); (2) **VRP$_{\mathbf{RL}}$** (RL under VRP); (3) **MRP$_{\mathbf{RL}}$** (RL under MRP with high-level task analysis, Eq. (4)), and (4) **ReMA** (ours, RL under MAMRP, Eq. (7)).

We train and test Llama-3-8B-Instruct, Llama-3.1-8B-Instruct [Dubey et al., 2024], and Qwen2.5-7B-Instruct [Team, 2024] on mathematical reasoning benchmarks. For LLM-as-a-judge benchmarks, we train and test Llama-3.1-8B-Instruct and Qwen2.5-7B-Instruct. We use instruct-tuned

---

[4]https://huggingface.co/datasets/AI-MO/aimo-validation-aime
[5]https://huggingface.co/datasets/AI-MO/aimo-validation-amc

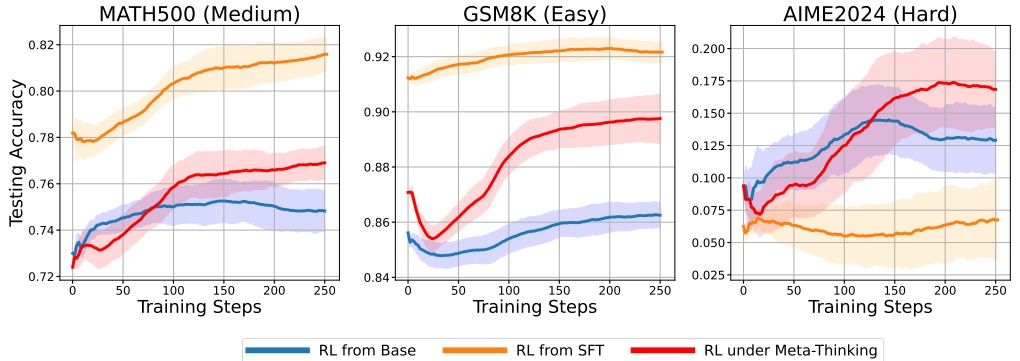

Figure 3: An RL experiment with 3 training schemes. While **RL from SFT** excels on easier problems, **RL under Meta-thinking** shows superior generalization to harder problems like AIME24.

LLMs to prompt them to perform VRP, MRP, and MAMRP directly during training. Unless specified, we use two separate copies of the same model for high- and low-level agents in ReMA. We use the base reward setting in Appendix C.2 by default. And for the underlying RL algorithm, we use REINFORCE++ [Hu, 2025]. We refer to Appendix D for detailed training settings.

## 4.2   Results of Single-turn ReMA

**Question 1.** *Does single-turn ReMA outperforms baselines on both in-distribution and out-of-distribution test sets?*

Table 1 compares the greedy decoding performance of ReMA against various RL baselines across mathematical benchmarks (Table 1a) and LLM-as-a-Judge benchmarks (Table 1b). Results across different LLMs indicate that, **on average, ReMA outperforms all baselines**, achieving a maximum improvement of 6.68% on mathematical benchmarks and 8.49% on LLM-as-a-Judge benchmarks.

Notably, ReMA achieves the highest performance on most benchmarks, particularly on out-of-distribution datasets, with a maximum improvement of 20% on AMC23 for Llama3-8B-Instruct, 13.33% on AIME24 for Qwen2.5-7B-Instruct, 14.23% on RewardBench970 for Llama3.1-8B-Instruct. These results demonstrate the superior out-of-distribution generalization ability conferred by the meta-thinking mechanism in ReMA. However, we observe that the accuracy gains from RL training on instruction-tuned LMs are smaller than from base models (Sec. 4.2.1). This may be due to the higher initial performance and the relatively fixed output distribution of instruction-tuned models, which limits the improvement and peak performance in RL.

We also compare ReMA with other inference-time meta-thinking and multi-agent baselines, e.g. Self-Refine[Madaan et al., 2023] and Multi-Agent Debate (MAD)[Du et al., 2023], we refer to Appendix E.1 for detailed results.

### 4.2.1   Meta-thoughts boost low-level generalization

**Question 2.** *Can Reasoning benefit from Meta-thinking?*

Here we provide a tiny but motivating example of how ReMA gives better learning dynamics. We use Qwen2.5-Math-7B [Yang et al., 2024] as the starting base model, MATH (level 3-5, about 5.5K number of instances) as the training dataset, and we compare three reinforcement learning training schemes, in particular: (1) **RL from Base**: train the base model directly on MATH with binary outcome reward; (2) **RL from SFT**: SFT the base model with GPT-4o's CoT answers; then RL on train dataset with binary outcome reward; (3) **RL under Meta-thinking**: SFT the base model with GPT-4o's meta-thinking plans; then RL on train dataset with binary outcome reward.

The models are evaluated on 3 benchmarks (Fig. 3). SFT brings the best initial accuracy on in-distribution and easier sets, but fails to improve on harder ones. **RL from Base** yields limited gains. In contrast, **RL under Meta-thinking achieves the best learning dynamics and generalizes better to challenging problems** (AIME24). See Appendix F.1 for case studies.

Table 1: Performance on in-distribution test sets and out-of-distribution test sets. We also report the improvement/degradation w.r.t. basic CoT performance(VRP). On average, ReMA outperforms all baselines. Particularly on out-of-distribution datasets, ReMA achieves the highest performance on most of the benchmarks.

(a) Performance on math benchmarks

| Model | Benchmark | VRP (CoT) | VRP$_{RL}$ | MRP$_{RL}$ | ReMA (Ours) |
|---|---|---|---|---|---|
| **Llama3 -8B -Instruct** | MATH500 | 30.80 | 33.40 (+2.60) | 32.80 (+2.00) | **33.80 (+3.00)** |
| | GSM8K | 67.48 | **81.80 (+14.32)** | 79.68 (+12.20) | 79.38 (+11.90) |
| | AIME24 | 0.00 | 0.00 (+0.00) | **3.33 (+3.33)** | 0.00 (+0.00) |
| | AMC23 | 2.50 | 10.00 (+7.50) | 12.50 (+10.00) | **22.50 (+20.00)** |
| | Gaokao2023en | 22.34 | 27.53 (+5.19) | 23.38 (+1.04) | **28.57 (+6.23)** |
| | Minerva Math | 8.82 | 16.54 (+7.72) | **18.01 (+9.19)** | 13.97 (+5.15) |
| | Olympiad Bench | 8.44 | 8.89 (+0.45) | **9.33 (+0.89)** | 8.89 (+0.45) |
| | Average | 20.05 | 25.45 (+5.40) | 25.58 (+5.53) | **26.73 (+6.68)** |
| **Llama3.1 -8B -Instruct** | MATH500 | 50.80 | 50.20 (-0.60) | 48.60 (-2.20) | **53.20 (+2.40)** |
| | GSM8K | 86.05 | 84.53 (-1.52) | 85.37 (-0.68) | **87.26 (+1.21)** |
| | AIME24 | 10.00 | 3.33 (-6.67) | 6.67 (-3.33) | **13.33 (+3.33)** |
| | AMC23 | 27.50 | 12.50 (-15.00) | **30.00 (+2.50)** | 20.00 (-7.50) |
| | Gaokao2023en | **38.96** | 36.10 (-2.86) | 37.14 (-1.82) | 37.14 (-1.82) |
| | Minerva Math | 22.79 | 26.84 (+4.05) | 25.37 (+2.58) | **28.31 (+5.52)** |
| | Olympiad Bench | 15.11 | **19.70 (+4.59)** | 15.70 (+0.59) | 19.56 (+4.45) |
| | Average | 35.89 | 33.32 (-2.57) | 35.55 (-0.34) | **36.97 (+1.08)** |
| **Qwen2.5 -7B -Instruct** | MATH500 | 75.00 | **77.20 (+2.20)** | 76.40 (+1.40) | 74.40 (-0.60) |
| | GSM8K | **92.04** | 91.36 (-0.68) | 91.81 (-0.23) | 90.60 (-1.44) |
| | AIME24 | 6.67 | 6.67 (+0.00) | 10.00 (+3.33) | **20.00 (+13.33)** |
| | AMC23 | 47.50 | 50.00 (+2.50) | 52.50 (+5.00) | **57.50 (+10.00)** |
| | Gaokao2023en | 56.62 | 54.81 (-1.81) | 55.06 (-1.56) | **57.92 (+1.30)** |
| | Minerva Math | **35.66** | 34.93 (-0.73) | 32.35 (-3.31) | 34.93 (-0.73) |
| | Olympiad Bench | 38.22 | **38.37 (+0.15)** | 37.78 (-0.44) | 36.30 (-1.92) |
| | Average | 50.24 | 50.48 (+0.24) | 50.84 (+0.60) | **53.09 (+2.85)** |

(b) Performance on LLM-as-a-Judge benchmarks

| Model | Benchmark | VRP (CoT) | VRP$_{RL}$ | MRP$_{RL}$ | ReMA (Ours) |
|---|---|---|---|---|---|
| **Llama3.1 -8B -Instruct** | RewardBench970 | 69.48 | 82.89 (+13.41) | 81.13 (+11.65) | **83.71 (+14.23)** |
| | JudgeBench | 51.29 | 51.94 (+0.65) | **52.90 (+1.61)** | **52.90 (+1.61)** |
| | Average | 60.39 | 67.41 (+7.02) | 67.02 (+6.63) | **68.31 (+7.92)** |
| **Qwen2.5 -7B -Instruct** | RewardBench970 | 78.56 | 85.36 (+6.80) | **86.49 (+7.93)** | 83.51 (+4.95) |
| | JudgeBench | **58.39** | 56.94 (-1.45) | 58.39 (+0.00) | 56.94 (-1.45) |
| | Average | 68.47 | 71.15 (+2.68) | **72.44 (+3.97)** | 70.22 (+1.75) |

#### 4.2.2 Diverse meta-thinking characteristics of LLMs

**Question 3.** *How well can LLMs learn to meta-think?*

To further analyze meta-thinking behaviors, we train models with structured JSON-format actions inspired by Yue et al. [2024]. The meta-thinking agent generate two entries in one LM call, first selects from three actions: DECOMPOSE (breaking into subproblems), REWRITE (simplifying the problem), or EMPTY (direct solving), then generates the corresponding text. We compare Llama-3.1-8B-Instruct and Llama-3.2-1B-Instruct to study scale effects (two 1B models vs two 8B models) on meta-thinking agent's training. We use vLLM guided JSON decoding [Dong et al., 2024] for valid formatting and base reward (reasoning agent's solution accuracy with format constraints).

We observe that smaller LMs produce simpler outputs, likely due to limited capacity to maintain valid JSON formatting while exploring diverse reasoning strategies. As Fig. 4 shows, smaller LMs like Llama-3.2-1B-Instruct quickly converge to the simplest EMPTY action to avoid format-

ting penalties, **while larger LMs like Llama-3.1-8B-Instruct can adapt meta-thinking strategies based on problem difficulty**. See Appendix F.3 for detailed case studies.

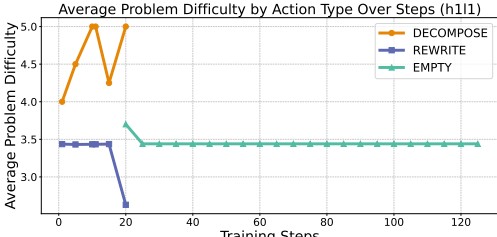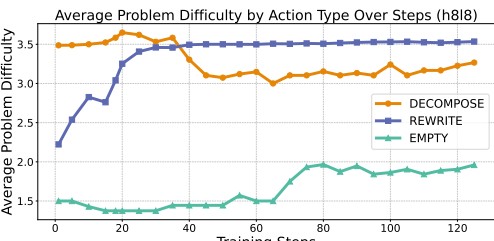

Figure 4: Average problem difficulty by action type during training. **Left**: 1B LM collapses to the `EMPTY` action. **Right**: 8B LM adapts to a more complex meta-thinking strategy for harder problems.

### 4.3 Extending ReMA to Multi-turn MAMRP

We further extend ReMA to multi-turn MAMRP settings, enabling multiple rounds of interaction between the meta-thinking agent and the reasoning agent as defined in Eq. (7).

Unlike the inherent VRP capabilities of most LLMs, multi-turn ReMA requires initial bootstrapping. Thus, we constructed a supervised fine-tuning dataset (about 0.8K samples) from LIMO [Ye et al., 2025c] using GPT-4o to establish the starting point for multi-turn interaction capabilities. Then we finetune Qwen2.5-7B before RL training.

As described in Sec.3.2.2, we deploy the proposed GRPO with turn-level ratio clipping and trajectory-level averaging loss during training. And we remove the KL-divergence term to allow more flexible exploration. By default, the agents share the same parameters and are simultaneously updated using their trajectories. We refer to details in Appendix D.2.

#### 4.3.1 Results and Ablations

**Question 4.** *Can ReMA be scaled to multi-turn settings?*

There are two key points revealed by our multi-turn ReMA experiments, as shown in Fig. 5. On one hand, **the algorithm can demonstrate effective convergence on the training set**, with accuracy steadily increasing from approximately 55% to 70% during training. It also achieves an average performance gain of about 5% across all seven test benchmarks, indicating stable improvements on out-of-distribution data. (Experiment with the rollout config of *turn30_token512*, see Appendix D.2.2 and Fig. 8 for more details.)

On the other hand, we observe that the performance of multi-turn ReMA is highly sensitive to hyperparameters such as the maximum response length per turn and the maximum number of turns. For certain configurations, the model either collapses into producing massive repetitions within a single turn or generates empty responses after only a few turns. Similar phenomena have been reported in concurrent works such as RAGEN [Wang et al., 2025] and nondeterminism between inference and training engines in current RL infra [He and Lab, 2025] and improper token processing that deviates from the token-in-token-out principle (e.g., performing risky decode-encode operations on tokens). As a result, multi-turn RL becomes susceptible to long-horizon credit assignment challenges and state drift, often leading to reduced exploration diversity—a phenomenon referred to as the "Echo Trap". To address this challenge, it is essential to comprehensively explore the training recipe w.r.t. model, data, algorithm and, most importantly, the **RL infra**.

**Question 5.** *How does parameter sharing and turn-level ratio affect multi-turn ReMA?*

As shown in Fig. 6, we compare different configurations on a smaller dataset consisting of 133 samples—19 from each of the 7 MATH problem types—to evaluate sample efficiency and convergence speed. We run each experiment with 3 different random seeds and report the average performance. First, all configurations eventually achieve nearly 100% accuracy on the training dataset. Notably, **the trajectory-level loss with turn-level ratio (*Turn-Ratio*, Eq. (14)) demonstrates substantially better sample efficiency** than its token-level variants (Eq. (2)), reaching higher training rewards with fewer steps. We also present the training curve of separate weight setting, the empirical results show that **shared parameters with simultaneous updates converge noticeably faster**.

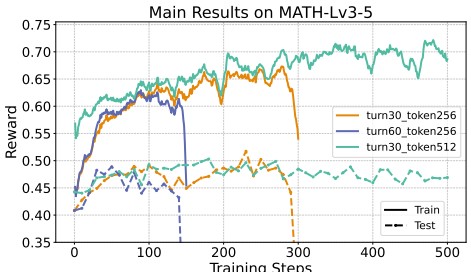
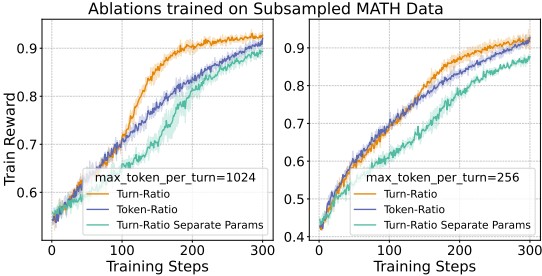

Figure 5: Training results of multi-turn ReMA on MATH-Level3-5-8K under different rollout configurations.

Figure 6: Ablations of multi-turn ReMA on a tiny subset of MATH, we only show here the training curves of different training and rollout configurations.

# 5 Conclusion

In this paper, we introduced ReMA, a novel framework that leverages multi-agent reinforcement learning to elicit meta-thinking in large language models. By explicitly separating meta-thinking and reasoning processes into distinct agents, our approach enhances both exploration during training and the interpretability of model outputs. We tailored RL algorithms and reward functions to ensure reliable performance. Through comprehensive experiments on mathematical reasoning and LLM-as-a-Judge benchmarks, ReMA consistently achieved superior results, particularly on out-of-distribution datasets. We further extend ReMA to multi-turn settings, enabling the framework to handle more complex reasoning scenarios that require more communication between agents. Our ablations demonstrate how effective coordination between agents evolves, highlighting the promise of reinforcement learning and structured agents' collaboration for advancing the capabilities of language models in complex reasoning tasks.

# 6 Acknowledgment

This work was supported by the Shanghai Municipal Science and Technology Major Project. The Shanghai Jiao Tong University team is supported by National Key R&D Program of China 2024YFC3505402, Doubao Large Model Fund. The Shanghai Jiao Tong University team is also partially supported by Shanghai Municipal Science and Technology Major Project (2021SHZDZX0102) and National Natural Science Foundation of China (62322603). This work was partially supported by the Canada CIFAR AI Chair Program, the Natural Sciences and Engineering Research Council of Canada (NSERC) Discovery Grants RGPIN-2022-03669, and enabled in part by support provided by the Digital Research Alliance of Canada (alliancecan.ca).

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

## Appendix Table of Contents

## A  Related work

Drawing from the bitter lesson [Sutton, 2019], two methods that appear to scale effectively are searching and learning, aligning with current trends in large language models [Xu et al., 2025]. At present, researchers are leveraging these methods to maximize the capabilities of individual transformers, while other efforts are exploring architectures that involve multiple interacting entities. In this paper, we examine this divergence within the context of LLM reasoning, a capability that allows large language models to solve problems through logical reasoning, step-by-step analysis, and inference [Wang et al., 2024a].

### A.1  Single LLM Reasoning

Main research works in reasoning involving a single LLM utilize search-based and post-training methods. The fundamental elements of searching methods are text generation and evaluation. Generation schemes include In-Context Learning [Brown et al., 2020], Beam Search [Graves, 2012], and various tree-based searching [Snell et al., 2024]; Evaluation approaches often use outcome accuracy, self-consistency [Wang et al., 2022], or process reward signal [Lightman et al., 2023] as the criteria to select high-quality responses from the generated texts. Post-training method is another research line in opposition to pre-training. Popular training pipelines often involve specific data

construction followed by Supervised Fine-tuning [Qin et al., 2024, Ouyang et al., 2022, Hui et al., 2024, Liu et al., 2024], or reinforcement learning to interactively explore learning patterns [Wang et al., 2024a, Zhang et al., 2024a, DeepSeek-AI et al., 2025, Xu et al., 2025].

## A.2   Multiple LLM Reasoning

Integrating multiple entities can potentially surpass the intelligence of the individual model [Chen et al., 2023]. With the rapid emergence of large language models showing a varying level of abilities, some studies have explored facilitating discussions among multiple off-the-shelf LLMs [Zhang et al., 2025a, Chen et al., 2024a, Wang et al., 2023, Du et al., 2023, Zhuge et al., 2023, Tang et al., 2023, Hao et al., 2025, Akata et al., 2023, Hong et al., 2023, Zhang et al., 2024b], taking the form of free discussion [Du et al., 2023, Liang et al., 2023] or structured role assignments [Hong et al., 2023, Zhang et al., 2024b]. Some have applied routing mechanisms to assign tasks to the most suitable expert models [Hu et al., 2024b, Stripelis et al., 2024, Ding et al., 2024, Yue et al., 2025a, Chen et al., 2024c] or merging mechanisms to develop more versatile models [Yadav et al., 2024, Yu et al., 2024, Zhang et al., 2025b]. Beyond aggregating static knowledge from multiple agents, multi-agent LLM training can also enhance reasoning capabilities. For example, multi-agent debates can generate diverse synthetic data, which can subsequently be used for supervised fine-tuning [Estornell et al., 2024, Li et al., 2024, Motwani et al., 2024, Dong and Ma, 2025, Perez et al., 2022, Ye et al., 2025a, Subramaniam et al., 2025]. Reinforcement learning (RL) methods have also been adopted to improve LLM reasoning in areas such as alignment [Perez et al., 2022, Ma et al., 2023] and legibility [Kirchner et al., 2024]. Motwani et al. [2024] utilize a three-agent system for generation and fine-tune the models using Direct Preference Optimization (DPO). Reinforcement Learning with Generative Reward Models (GenRM) [Mahan et al., 2024, Ye et al., 2025b, Jiao et al., 2024, Wang et al., 2024b] represents another common approach of multi-agent training, where the reward signal is derived from the token probabilities of another LLM, coupled with the reasoning process. While our work aligns with these efforts, it diverges by using an additional tunable LLM to provide metacognitive instructions, guiding the low-level LLM during learning, rather than relying on a static GenRM. The most closely related works to ours are MAPoRL [Park et al., 2025] and COPRY [Ma et al., 2024]. MAPoRL is a multi-agent debating framework that uses multi-agent reinforcement learning (MARL) with a learned verifier to fine-tune each LLM agent. COPRY duplicates an LLM into two agents, training them simultaneously in the roles of pioneer and observer using RL. Shen et al. [2025] trained with a novel Chain-of-Action-Thought (COAT) framework that embeds meta-action tokens for self-reflection and exploration into an autoregressive search process. However, unlike our approach, which explicitly separates metacognition from plan execution, these methods do not decompose the reasoning process but instead focus on improving direct chain-of-thought generation. Furthermore, our experiments are conducted on a larger scale and include more challenging problems.

## A.3   Hierarchical Reasoning

Partitioning reasoning into hierarchical processes has been explored in prior research to make biological sense [Ye et al., 2018, Langley et al., 2004]. In the context of language models, a hierarchical structure has been used to facilitate diverse reasoning patterns, including planning [Puerta-Merino et al., 2025, Sun et al., 2024, Song et al., 2023, Rana et al., 2023, Chen et al., 2024d, Yan et al., 2023, Xiao et al., 2024], validation [Haji et al., 2024, Xi et al., 2024] and self-refinement [Madaan et al., 2023, Kumar et al., 2024, Welleck et al., 2022]. For instance, EvalPlanner [Saha et al., 2025] is a framework that conducts reasoning through plan generation and execution. DOTS [Yue et al., 2024] extends decomposition by integrating a tree-based searching method with Analysis, Solution, and Verification layers. Marco-o1 [Zhao et al., 2024] focuses on open-ended problem-solving and abstract thinking, dynamically adjusting reasoning granularity and incorporating reflection mechanisms to enhance reasoning performance. Beyond these approaches, metacognition [Flavell, 1979] has been identified as another critical component of reasoning, referring to the intuitive understanding of one's own cognitive and reasoning processes [Gao et al., 2024, Wang and Zhao, 2023]. Wang and Zhao [2023] proposed a metacognitive prompting strategy to improve large language model (LLM) capabilities. Didolkar et al. [2024] further developed a prompt-guided method that enables models to label math problems with the required skills and subsequently use these labels to solve new problems. Gao et al. [2024] introduce meta-reasoner which use contextual multi-arm bandit to learn a high-level "advisor" over low-level reasoning process. Xiang et al. [2025] provides a

Meta-CoT framework to think about its own thinking. They use construction-based methods as well as reinforcement learning to develop meta-cognitive skills. Qingsong et al. [2025] introduces a RL framework for dynamic instruction selection during fine-tuning. In our work, we also value reflecting on reasoning processes, and we enhance metacognitive abilities through two-agent interaction and reinforcement learning at both end.

### A.4 RL in LLM

Recent advancements in applying RL to LLMs have enhanced their reasoning and decision-making capabilities. Liu et al. [2025] examines token-level optimization biases by introducing Dr. GRPO to stabilize policy gradients. VAPO [Yue et al., 2025b] enhances PPO with value-aware perturbations and adaptive reward shaping to improve robustness in sparse-reward reasoning tasks. DAPO [Yu et al., 2025] provides a scalable, modular RL framework that integrates distributed rollout collection and dynamic replay buffers for reproducible training at scale. SimpleRL-Zoo [Zeng et al., 2025] conducts zero-shot RL experiments across open-base LLMs to uncover emergent cognitive behaviors under minimal reward signals. Echo Chamber [Zhao et al., 2025] investigates how RL fine-tuning algorithms can amplify pretrained model biases and proposes regularization to mitigate over-amplification. Wen et al. [2024] decomposes high-level language actions into token-level operations to achieve finer-grained credit assignment. Some works push RL training for single-turn to multi-turn. Search-R1 [Jin et al., 2025] trains LLMs to orchestrate multi-turn search strategies with RL-optimized decision policies to improve question-answering accuracy. ArCHer [Zhou et al., 2024] employs a hierarchical, multi-turn RL architecture with manager and worker policies to efficiently handle long-horizon dialogue tasks. RAGEN [Wang et al., 2025] introduces trajectory filtering and critic modules within a multi-turn RL framework to stabilize learning and reduce shallow policy behaviors.

## B Limitation and Future Work

In this work, we only test ReMA on math and LLM-as-a-Judge benchmarks. Though the results shows the effectiveness of ReMA, adopting ReMA to tasks where naturally needs multi-turn interaction between several interleaved agents has great potential. Moreover, a comprehensive understanding of the learning dynamics of multi-turn RL and multi-turn MARL for LLMs is needed. Finally, there's still sufficient space to further improve the procedure of multi-turn multi-agent rollout through modern LLM speed up techniques, e.g. prefill-decode disaggregation and asynchronous rollout.

## C Supplementary Materials for Method in Section 3

### C.1 Inference-time Scaling of ReMA

In this section, we discuss how to enhance the inference-time computation of our hierarchical system, specifically focusing on the interaction between the high-level and low-level agents. The total number of model samples required for inference is determined by the product of the sampling budget allocated to each agent.

For instance, in a simple single-turn setting, if the high-level agent samples $k_1$ responses and each of these responses leads to $k_2$ samples from the low-level agent, the total number of model calls required is:

$$\text{Total samples} = k_1 \times k_2.$$

Given a fixed computational budget, an important question arises: **how should the sampling budget be distributed between the high-level and low-level agents to maximize performance?** Allocating more samples to the high-level agent may increase diversity in reasoning strategies while allocating more to the low-level agent may yield more refined solutions for a given metacognitive plan.

Another crucial consideration is **how to perform reranking on the final outputs**. Two potential strategies include:

- **Hierarchical reranking:** First, for each high-level response, rank and aggregate the low-level responses under it. Then, rank the aggregated results across different high-level responses.
- **Flat reranking:** Directly rank all sampled responses together, regardless of the hierarchy of high-level reasoning steps.

Balancing sampling allocation and designing an effective reranking strategy are key challenges in efficiently scaling our multi-agent reasoning system. In the next section, we explore empirical results comparing different allocation strategies and ranking methods.

## C.2 Detailed reward design

As described in Sec. 3.2, we update both high-level and low-level agents by assigning rewards based on the low-level policy output. Below, we outline several potential reward designs:

- Correctness reward: For tasks with explicit ground truth, we assign rewards based on the correctness of the low-level agent's output.
- Format reward: For tasks that require a specific output format, we enforce adherence to the prescribed structure by providing a format reward.
- To encourage the high-level agent to generate informative and unambiguous meta-thinking, and to stabilize the low-level outputs, we reward the high-level agent when the low-level agent produces consistent responses. Specifically, the consistency reward is defined as

$$R_h = \frac{\text{max occurrence of an answer}}{\text{total number of responses}}.$$

To examine multi-agent metacognition-integrated reasoning with different reward designs, we experiment with different reward function designs to encourage effective collaboration and structured reasoning. Below, we introduce and justify several reward schemes.

**1. Correctness and Format-Aware Reward (Base Setting)** In our primary reward setting, the system's overall correctness is used as the primary reward signal, supplemented by format-based rewards for both the high-level and low-level agents. Using mathematical problem-solving as an example:

- **Low-level agent** ($\pi_{\theta_l}$): Receives a reward of $+1.0$ for a correct answer. If the answer is incorrect, the agent is further penalized based on format compliance. Specifically:
    - If the output contains the designated answer-indicating format (e.g., `boxed` in La-TeX), it receives $-0.5$.
    - Otherwise, it receives $-1.0$, as a missing format often suggests an incomplete or unstructured response.
- **High-level agent** ($\pi_{\theta_h}$): Receives the average correctness of the low-level agent's sampled responses as its reward. Additionally, to prevent the high-level agent from directly generating explicit answers instead of guiding reasoning, a strong penalty of $-1.0$ is applied if it includes an explicit answer format (e.g., `boxed`).

**2. Consistency-Based Reward** Instead of using correctness as the high-level reward signal, this approach rewards the high-level agent for promoting consistent responses from the low-level agent, regardless of actual correctness. The consistency reward is defined as the proportion of the most frequently occurring answer among all sampled responses:

$$R_h = \frac{\text{max occurrence of an answer}}{\text{total number of responses}} \tag{15}$$

If the majority of responses do not contain a definitive answer, the reward is set to zero. We also add the format penalty to the high-level agent if its output contains the designated answer-indicating format. This incentivizes the high-level agent to guide the low-level agent toward more stable, detailed, reproducible outputs rather than erratic reasoning paths.

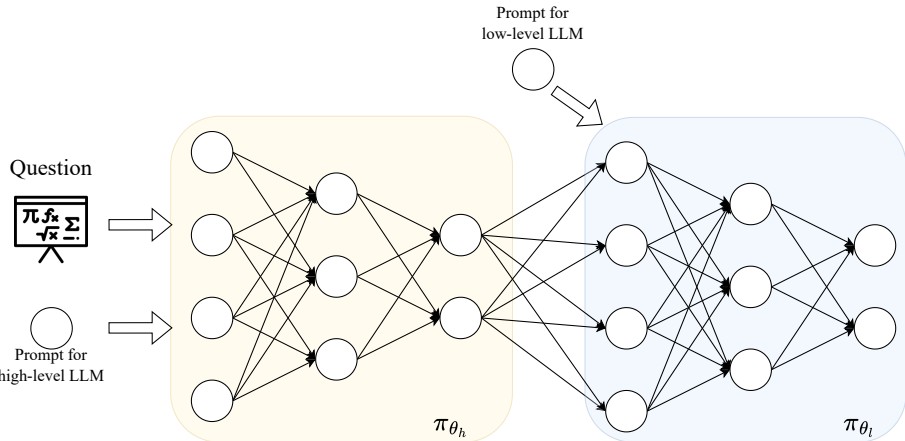

Figure 7: Our method can be viewed as a combination of practical TRPO and block coordinate ascent, with the high and low-level models treated as distinct components within a larger neural network. Note that the figure does not represent the exact gradient back-propagation flow but rather highlights the key idea that we separate the high- and low-level models. This separation allows for the independent computation of gradients and the independent training of each model.

These different reward formulations allow us to investigate various dimensions of metacognitive reasoning: correctness, consistency, etc. We empirically compare their effects on learned metacognitive reasoning patterns in Sec. E.2.

## C.3 Pseudocode of ReMA

The pseudocode is shown in Algorithm 1.

---

**Algorithm 1** Single turn MAMRP

---

**Require:** High-level policy $\pi_h$, Low-level policy $\pi_l$, Dataset $\mathcal{D}$, Optimizers for $\pi_h$ and $\pi_l$. $\varepsilon_{\min}, \varepsilon_{\max}$ to filter training dataset
1: Initialize $\pi_h$ and $\pi_l$
2: **while** not converged **do**
3:     build training dataset $\mathcal{D}_l$ with $\pi_h, \pi_l, \varepsilon_{\min}, \varepsilon_{\max}$
4:     **for** Sample $(\mathbf{x}, \mathbf{m}, \mathbf{y}^*) \sim \mathcal{D}_l$ **do**
5:         Generate $\mathbf{y} \sim \pi_l(\mathbf{x}, \mathbf{m})$
6:         Compute low-level reward $R_l(\mathbf{y}, \mathbf{y}^*)$
7:         Update $\pi_l$ using $\nabla_{\theta_l}\mathbb{E}[R_l]$
8:     **end for**
9:     build training dataset $\mathcal{D}_h$ with $\pi_h, \pi_l, \varepsilon_{\min}, \varepsilon_{\max}$
10:     **for** Sample $(\mathbf{x}, \mathbf{y}^*) \sim \mathcal{D}_h$ **do**
11:         Generate $\mathbf{m} \sim \pi_h(\mathbf{x})$ and $\mathbf{y} \sim \pi_l(\mathbf{x}, \mathbf{m})$
12:         Compute high-level reward $R_h(\mathbf{m}, \mathbf{y}, \mathbf{y}^*)$
13:         Update $\pi_h$ using $\nabla_{\theta_h}\mathbb{E}[R_h]$
14:     **end for**
15: **end while**

---

## C.4 Brief convergence analysis

We reuse the notations from Sec. 3.2, where $\mathbf{x}$ is task prompt, $\mathbf{y}$ is generated answer, $\mathbf{y}^*$ is ground-truth, $\mathbf{m}$ is metacognition on task solving, $\pi_{\theta_h}$ and $\pi_{\theta_l}$ are high- and low-level agents with parameters $\theta_h$ and $\theta_l$. We consider the joint hierarchical policy defined in Eq. (8) and update the objective as in Eq. (9).

To leverage existing RL and optimization convergence analysis methods, we treat the two models as components of a larger model, as illustrated in Fig. 7. When updating one model, we treat the other model as part of a stationary environment. The gradients with respect to $\theta_h$ and $\theta_l$ are:

$$\nabla_{\theta_h} J(\theta_h, \theta_l) = \mathbb{E}_{\mathbf{x}, \mathbf{y}^*} \sum_{\mathbf{m} \sim \pi_h(\mathbf{m}|\mathbf{x}; \theta_h)} \nabla_{\theta_h} \pi_h(\mathbf{m} \mid \mathbf{x}; \theta_h) \left[ \mathbb{E}_{\mathbf{y} \sim \pi_l(\mathbf{y}|\mathbf{x}, \mathbf{m})} R(\mathbf{y}, \mathbf{y}*) \right],$$

$$\nabla_{\theta_l} J(\theta_h, \theta_l) = \mathbb{E}_{\mathbf{x}, \mathbf{y}^*} \sum_{\mathbf{y} \sim \pi(\theta_h, \theta_l)} \nabla_{\theta_l} \pi_l(\mathbf{y} \mid \mathbf{x}, \mathbf{m}; \theta_h); \theta_l) R(\mathbf{y}, \mathbf{y}*).$$

We can compute the gradients with log trick and estimate $\mathbb{E}_{\mathbf{y} \sim \pi_l(\mathbf{y}|\mathbf{x}, \mathbf{m})} R(\mathbf{y}, \mathbf{y}*)$ with Monte Carlo method.

Equipped with the objective function and gradient computation, we update the models iteratively. Without loss of generality, we analyze the case where the high-level policy is updated first:

$$\theta_h^{(t+1)} = \arg\max_{\theta_h} J(\theta_h, \theta_l^{(t)}),$$

$$\theta_l^{(t+1)} = \arg\max_{\theta_l} J(\theta_h^{(t+1)}, \theta_l).$$

Regarding the different regularizations $R_h$ and $R_l$ in Eqs. (10) and (11) for the different policies, instead of directly integrating them into the loss function, we treat them as constraints, as done in Trust Region Policy Optimization (TRPO) [Schulman et al., 2015]. Note that when one policy is fixed, the other policy operates in a stationary decision process.

Based on the defined objective and update method, we apply TRPO and block coordinate ascent. First, recall that when updating a single policy, TRPO guarantees monotonic improvement by optimizing a lower bound. Specifically, let $\pi_{\text{old}}$ and $\pi$ represent the old and current policies, respectively. We define a surrogate objective as:

$$L_{\pi_{\text{old}}}(\pi) = \mathbb{E}_{s \sim \pi_{\text{old}}, a \sim \pi_{\text{old}}} \left[ \frac{\pi(a|s)}{\pi_{\text{old}}(a|s)} A^{\pi_{\text{old}}}(s, a) \right],$$

As shown by Schulman et al. [2015], the true objective of $\pi$ is lower-bounded by:

$$J(\pi) \geq L_{\pi_{\text{old}}}(\pi) - C \cdot \max_s \text{KL}[\pi_{\text{old}}(\cdot \mid s), \pi(\cdot \mid s)],$$

for some constant $C$. By optimizing the right-hand side of the above inequality, we are guaranteed to improve the performance of $\pi$. Therefore, for policies $\pi^t$ and $\pi^{t+1}$ obtained from iterations $t$ and $t + 1$ using the TRPO method, we have:

$$J(\pi^{t+1}) \geq J(\pi^t).$$

Now, returning to our updating method, we treat the high- and low-level policies as two blocks of a single agent. The iterative update process can thus be viewed as a cyclic block coordinate ascent, where the two policies are updated in a fixed order. By updating each block using the TRPO method, and improving the surrogate objective within the KL constraint, each block update does not decrease $J$:

$$J(\theta_h^{t+1}, \theta_l^t) \geq J(\theta_h^t, \theta_l^t),$$
$$J(\theta_h^{t+1}, \theta_l^{t+1}) \geq J(\theta_h^{t+1}, \theta_l^t).$$

Thus $J(\theta_h^{t+1}, \theta_l^{t+1}) \geq J(\theta_h^t, \theta_l^t)$. This repeated coordinate maximization converges to a fixed point, where no single coordinate update can further improve $J(\theta_h, \theta_l)$.

Given the theoretical monotonic improvement with TRPO and block coordinate ascent, we adopt a practical version of TRPO in our experiments, specifically Proximal Policy Optimization (PPO) [Schulman et al., 2017] or GRPO [Shao et al., 2024].

### C.5 Learning to reason from the perspective of Leader Follower Game

Besides the loss function in the main part, we also propose to frame the problem as a leader-follower game. By analyzing the equilibria of the leader-follower game, we demonstrate that our framework inherently identifies the optimal sub-tasks aligned with the capabilities of the low-level model. This ensures that the high-level decisions are guided by the low-level model's strengths, leading to more efficient and targeted task decomposition.

### C.5.1 Leader-follower game

The leader-follower game, also known as the Stackelberg game, models interaction between two agents with parametrized strategies $\boldsymbol{\theta} = (\boldsymbol{\theta}_1, \boldsymbol{\theta}_2)$ and differentiable objective functions $(\mathcal{L}_1, \mathcal{L}_2)$ : $\mathbb{R}^d \to \mathbb{R}$. In this framework, the leader announces its strategy first, and the follower observes this decision to respond optimally. This sequential structure enables the leader to anticipate the follower's reaction and adjust its strategy accordingly. A Stackelberg equilibrium occurs when neither agent can unilaterally improve its objective. Denoting $\boldsymbol{\theta}_1$ as the leader's strategy and $\boldsymbol{\theta}_2$ as the follower's, the loss functions $\mathcal{L}_1$ and $\mathcal{L}_2$ are optimized with the following bi-level structure:

$$\boldsymbol{\theta}_1^* = \operatorname{argmin}_{\boldsymbol{\theta}_1} \mathcal{L}_1(\boldsymbol{\theta}, \boldsymbol{\theta}_2^*(\boldsymbol{\theta}_1)), \quad \boldsymbol{w}_2^*(\boldsymbol{\theta}_1) = \operatorname{argmin}_{\boldsymbol{\theta}_2} \mathcal{L}_2(\boldsymbol{\theta}_1, \boldsymbol{\theta}_2).$$

Anil et al. [2021] apply the leader-follower game to ensure checkable answers in a prover-verifier game (PVG). The objective is a verifier that is both complete (accepts all correct proofs from a verifier) and sound (rejects all incorrect proofs from a verifier). They analyze different scenarios where the verifier acts as the leader, the prover as the follower, and both announce strategies simultaneously, forming a Nash equilibrium. The study concludes that in verifier-led PVG, a Stackelberg equilibrium is both necessary and sufficient for achieving a sound and complete verifier, whereas in other configurations, a Stackelberg equilibrium is not necessary or sufficient for this outcome.

### C.5.2 Efficacy of LLM

Because the high-level policy possesses strong generalization capabilities, it is impractical for it to exhaustively explore every potential sub-task for each question. Instead, it naturally focuses on tasks within a feasible range of difficulty, leveraging only a limited set of coarse planning actions. Rather than pinpointing perfectly tailored sub-tasks, the policy searches for general tasks of particular computational complexity, *i.e.*, difficulty, that it can handle reliably. Motivated by this perspective, we incorporate the concept of a reasoning boundary for large language models (LLMs) [Chen et al., 2024b]. Intuitively, the reasoning boundary circumscribes the maximum difficulty of problems a model can solve at a desired accuracy level. Formally, for a model $\theta$, a task $t$, and a predefined threshold $A$, the reasoning boundary of $\theta$ represents the maximum problem difficulty $d$ that satisfies:

$$\mathcal{B}_{Acc=A}(t|\theta) = \sup_{d}\{d|Acc(t|d,\theta) = A\}.$$

where $d$ denotes the problem difficulty. By quantifying the difficulty level a model can reliably handle, the reasoning boundary provides a systematic way to align the high-level policy's focus with the model's actual capabilities, gauge the efficacy of the low-level policy, and determine the optimal strategy for solving the question.

### C.5.3 Leader-follower Game for LLM Reasoning

Our goal is to find the high-level policy that searches for the sub-task sequence based on the efficacy of the low-level policy to solve the question. We design the loss functions as follows:

$$\mathcal{L}_h = \mathbb{E}_{(x,y) \sim p_D, t_{1:K}} \left[ -\log \pi_l(y_K \mid x, t_{1:K}, y_{1:K-1}) \right],$$
$$\mathcal{L}_l = \mathbb{E}_{x \sim p_D, t_{1:k} \sim \pi_h, \hat{y}_k \sim \pi_l} \left[ -r(y_k, \hat{y}_k \mid x, t_{1:k}, y_{1:k-1}) \right],$$

where $r(y_k, \hat{y}_k \mid x, t_{1:k}, y_{1:k-1})$ represents the step reward for the correctness of $\hat{y}_k$ derived from the question $x$, the sub-task sequence $t_{1:k}$ from the high policy and prior intermediate answer $y_{1:k-1}$. The loss functions can be interpreted as follows: the high-level policy is incentivized to find sub-tasks that lead to the correct answer based on the capabilities of the low-level policy, while the low-level policy is incentivized to enhance its instruction-following ability.

How to minimize the loss functions and whether such minimization leads to the desired results remain questions. To explore this, we consider a simplified case of our method, where the high-level policy plans the complete sub-task sequence at the beginning and the low-level executes the instruction in a single interaction. The corresponding parameterized policies are defined as $\pi_h((t_1, \ldots, t_K) \mid x)$ and $\pi_l((\hat{y}_1, \ldots, \hat{y}_K) \mid x, (t_1, \ldots, t_K))$. The corresponding loss functions are:

$$\mathcal{L}_h = \mathbb{E}_{(x,y) \sim p_D, t_{1:K}} \left[ -\log \pi_l(y_K \mid x, t_{1:K}) \right], \tag{16}$$
$$\mathcal{L}_l = \mathbb{E}_{x \sim p_D, t_{1:k} \sim \pi_h, \hat{y}_k \sim \pi_l} \left[ -r(y_k, \hat{y}_k \mid x, t_{1:k}, y_{1:k-1}) \right]. \tag{17}$$

In this step, the high-level policy generates the entire sub-task sequence without relying on intermediate answers, while the low-level policy follows the sequence to produce answers for the sub-tasks. The low-level policy can still leverage prior intermediate answers to sequentially refine its responses.

To analyze the result agents by minimizing the loss functions, we adopt the completeness and soundness properties from the PVG framework for LLM reasoning. Specifically, if the high-level policy generates a sub-task sequence that is executable within the low-level policy's capabilities, the problem must be solved (completeness). Conversely, if the sub-task sequence is incorrect or beyond the low-level policy's capacity, the problem cannot be solved (soundness). To achieve this, we utilize the conclusion from Anil et al. [2021], which positions the low-level policy as the leader and the high-level policy as the follower, equilibria guarantee the complete and sound low-level policy.

When the high-level policy takes the lead, the low-level policy is forced to adapt to the specific strategy defined by the high-level policy, which can result in neither complete nor sound low-level policy. For example, if the high-level policy dictates that it will only generate sub-tasks involving addition and subtraction, the low-level policy is constrained to optimize only for these tasks. While they may reach an equilibrium, the low-level policy remains incomplete, and this limitation impacts both policies. In the case of the simultaneous PVG game, convergence to a Nash equilibrium is possible, but it is not sufficient for completeness and soundness. For instance, the low-level policy might disregard the high-level policy entirely (e.g., if the high-level provides incorrect instructions, but the low-level still performs correctly). This approach, however, is challenging to implement due to the significantly larger search space involved.

Furthermore, the loss functions we design ensure that, at a Stackelberg equilibrium, the high-level policy identifies sub-task sequences that the low-level policy can execute to solve the problem with the highest probability. With the low-level policy acting as the leader, it establishes its reasoning boundary for tasks. Based on the reasoning boundary, let $\theta_h$ and $\theta_l$ represent the policy parameters for the high-level and low-level policies, respectively. The probability that the low-level policy correctly solves the question is defined as:

$$\pi_l(y_K \mid x, t_{1:K}) = \prod_{k=1}^{K} \mathrm{Acc}(t_k \mid x, \theta_l),$$

where we can compute the difficulty $d_k$ from $t_k$ and $x$. where the difficulty $d_k$ can be derived from $t_k$ and $x$. The loss function in Eq. (16) ensures that the selected sub-tasks are optimal for the low-level policy. Here we provide a theoretical condition under which the most efficient solution strategy can be identified, according to the efficacy of the LLM.

This approach can be viewed as a game between a high-level "prover" and a low-level "verifier". The verifier, representing the low-level policy, adheres the high-level policy's instructions to validate its reasoning. Unlike the classic PVG setting, where the prover has ground-truth labels, the label of our high-level policy depends on the tunable low-level policy. This distinction, where the low-level policy (leader) is inherently more complex, contrasts with traditional PVG setups and adds complexity due to the interdependence between the high- and low-level policies.

By framing the problem-solving process as a leader-follower game, with the low-level policy designated as the leader, we can construct a bi-level optimization problem to identify an equilibrium. Following the formulation in Sec. C.5.1, the problem is expressed as:

$$\theta_l^* = \arg\min_{\theta_l} \mathcal{L}_l(\theta_h^*(\theta_l), \theta_l) \quad \theta_h^*(\theta_l) = \arg\min_{\theta_l} \mathcal{L}_h(\theta_h, \theta_l).$$

Then we can apply bi-level optimization techniques.

# D  Training Details

## D.1  Single-turn ReMA

We refer to Appendix G for prompts we use during training. We implement the training pipeline with OpenRLHF [Hu et al., 2024a] which is a highly efficient codebase and is easy to scale up. We select REINFORCE++ to save resources and for efficient training. All experiments are conducted in a node of 8 NVIDIA A100 GPUs. We use bf16, Zero2, Flash-Attention and gradient checkpointing to run our experiments.

During rollout, we set temperature=1.0, top_p=1.0, top_k=-1, and use vLLM for inference acceleration. We set the max generation length to be 2048 and, the rollout batch size to be 1000. The number of samples per prompt is 4. During training, we use Adam Optimizer with a learning rate of 5e-7. We set the mini-batch size to be 500, and the clip ratio to be 0.2. Other hyperparameters, such as KL coefficients and the number of training episodes, were carefully tuned based on validation set performance to ensure robust and reliable results. To align with the hyperparameter in OpenRLHF, we use #Training Episode as the number of reinforcement learning epoch on the entire dataset.

In ReMA, during prompt filtering of the high-level model, the high-level agent first samples 10 candidates for each question with t=1.0, and for each output the low-level agents sample 1 solution with t=0.0, then we select questions of success rate between $[\varepsilon_{\min}, \varepsilon_{\max}]$. And for the low-level agent's prompt filtering, the high-level agent first samples 1 candidate for each question with t=0.0, and for each output the low-level agents sample 10 solutions with t=1.0, then we select questions of success rate between $[\varepsilon_{\min}, \varepsilon_{\max}]$ and use the high-level agent to sample 4 meta-thoughts with t=1.0 as the input.

### D.1.1 Supervised fine-tuning data collection

For experiments in Sec. 4.2.1, we collect expert data to enhance the reasoning pattern, *i.e. RL from SFT*. Specifically, we collect demonstration data from GPT-4o Mini on MATH training dataset (7.5k problems) Hendrycks et al. [2021] and use it to fine-tune the LLMs. The data generation follows these steps: First, we prompt GPT-4o Mini to produce metacognitive reasoning for high-level model training. Specifically, we use different prompts to instruct it to rewrite and decompose a given question without providing a final answer. We collect metacognitive reasoning using two predefined actions, "rewrite" and "decompose", which align with human approaches to complex problem-solving while preserving answer diversity. Next, we use the generated instructions to prompt GPT-4o Mini to follow the metacognitive steps and solve the question, obtaining SFT data for low-level policy training. Below, we present the prompts used for both high-level and low-level models. Prompts can be found in Appendix G.1.1.

### D.1.2 Dataset Curation of RewardBench970

Table 2: Performance on LLM-as-a-Judge benchmarks, trained on dataset under the loose setting. The two-agent workflow in ReMA

| Model | Benchmark | VRP (CoT) | VRP$_{\text{RL}}$ | MRP$_{\text{RL}}$ | ReMA (Ours) |
|---|---|---|---|---|---|
| **Llama3.1 -8B -Instruct** | RewardBench970 | 71.24 | 81.86 (+10.62) | 80.41 (+9.17) | **86.29 (+15.05)** |
| | JudgeBench | 51.77 | 51.45 (-0.32) | 50.65 (-1.12) | **53.71 (+1.94)** |
| | Average | 61.51 | 66.65 (+5.14) | 65.53 (+4.02) | **70.00 (+8.49)** |
| **Qwen2.5 -7B -Instruct** | RewardBench970 | 86.49 | 87.22 (+0.73) | 80.31 (-6.18) | **90.72 (+4.23)** |
| | JudgeBench | 58.39 | 54.84 (-3.55) | 55.81 (-2.58) | **58.71 (+0.32)** |
| | Average | 72.44 | 71.03 (-1.41) | 68.06 (-4.38) | **74.72 (+2.28)** |

We process the original dataset in RewardBench by splitting it into a training set containing 5,000 tuples of (instruction, response A, response B) and a test set with the remaining 970 tuples.

To ensure a meaningful dataset split, we validate two separation strategies:

- Loose setting: We only ensure that there is no direct overlap of tuples between the training and test sets.
- Strict setting: We further enforce that no instruction appears in both the training and test sets. The results for this setting are presented in the main results (Table 1b).

Additionally, since the original RewardBench data originates from different subsets, we ensure that all original subsets are evenly represented in both the training and test sets.

Table 2 reports the learning performance of various methods under the loose dataset split setting. Compared to the results in Table 1b, ReMA significantly outperforms other RL tuning baselines

across all models, particularly on out-of-distribution (OOD) benchmarks. **The consistent improvements on OOD datasets of these two settings suggest that ReMA enhances meta-thinking ability, resulting in better generalization across diverse task distributions.**

### D.1.3 Training on MATH

**VRP** For Llama3-8B-Instruct, Llama3.1-8B-Instruct, and Qwen2.5-7B-Instruct, we all use a KL coefficient of 1e-2, and for #Training Episode, we use 12,6,6 for these 3 models respectively. For Llama3-8B-Instruct, we set the learning rate of 2e-7 for stable training.

**MRP** For Llama3-8B-Instruct, Llama3.1-8B-Instruct, and Qwen2.5-7B-Instruct, we all use a KL coefficient of 1e-2, and for #Training Episode, we use 10,6,6 for these 3 models respectively.

**MAMRP** We use $\varepsilon_{\min} = 0.2, \varepsilon_{\max} = 0.8$ for prompt filtering. We use the same #Training Episode=4 for all models, and for #Update Iteration, we use 3 for Llama3-8B-Instruct and Llama3.1-8B-Instruct, 10 for Qwen2.5-7B-Instruct. And we set the KL coefficient to be 1e-2 for all the 3 models.

### D.1.4 Training on Reward Bench

**VRP** For Llama3.1-8B-Instruct, and Qwen2.5-7B-Instruct, we all use a KL coefficient of 1e-2, and for #Training Episode, we use 4,6 for these 2 models respectively.

**MRP** For Llama3.1-8B-Instruct, and Qwen2.5-7B-Instruct, we all use a KL coefficient of 1e-2, and for #Training Episode, we use 4,6 for these 2 models respectively.

**MAMRP** We set #Update Iteration=1 for all models. We set the KL coefficient to be 1e-2 for Llama3.1-8B-Instruct and 1e-2 for Qwen2.5-7B-Instruct all models. For Llama3.1-8B-Instruct, we use $\varepsilon_{\min} = 0.2, \varepsilon_{\max} = 0.8$ for prompt filtering and we use #Training Episode of 2 during training. For Llama3.1-8B-Instruct, we use $\varepsilon_{\min} = 0.1, \varepsilon_{\max} = 0.9$ for prompt filtering and we use #Training Episode of 1 during training.

### D.2 Multi-turn ReMA

We refer to Appendix G for prompts we use during training. We implement a multi-turn ReMA training pipeline with VeRL [Sheng et al., 2024] since it's easier to implement complex training pipeline with a single centralized controller. Similar to OpenRLHF, VeRL is also a highly efficient and scalable codebase for further development.

For the multi-turn ReMA rollout, we use parameter sharing and simultaneous update by default. In details, we maintain two message lists with the system prompt of meta-thinking agent and reasoning agent respectively. During rollout, each agent act as 'assistant' in its own message list and the other agent act as 'user'. We use three hyperparameters to control the rollout length: (1) 'max_num_turns': the maximum number of turns for each trajectory. (2) 'max_response_length': the maximum number of tokens for each turn's response. (3) 'max_prompt_length': the maximum number of tokens for each trajectory.

During training, we apply the collected message list to Qwen2.5-7B's chat template and build loss masks in order to compute the loss for all turns of one trajectory (message list).

Morever, for multi-turn ReMA rollout, unlike single agent single turn rollout, we need to carefully design the termination logic. Basically, **we let the meta-thinking agent automatically decides when to finish the solving procedure**, we use a special tag '[FINISH]' to indicate the end of the solving procedure. After we detect this tag, we will terminate trajectory after the reasoning agent generates its output.

We also design other termination conditions to ensure the quality of the generated trajectories. If the last agent's response is too long, we will terminate the whole trajectory and setting the reward to 0. We also introduce a different version of format reward: we give a reward of 1.0 only if the reasoning agent's last turn response is correct and the meta-thinking agent's last turn response include '[FINISH]'. We use math_verify as the default verifier.

### D.2.1 SFT data collection of multi-turn MAMRP

We use GPT-4o to translate 817 samples in LIMO [Ye et al., 2025c] by prompting it to wrap each sentence with meta-thinking and reasoning tags. We use a temperature of 0. After filtering, we get 800 conversations for training. The prompt can be found in Appendix G.2.1. For supervised fine-tuning, we use LlamaFactory as the codebase and train the model for 3 epochs with a learning rate of 1e-5, consine learning rate scheduler, and batch size of 8. Use DeepSpeed Zero2 for distributed training.

### D.2.2 Training on MATH

For training of multi-turn ReMA on MATH, we use GRPO [Shao et al., 2024] as the default learning algorithm. We refer to Appendix G.2.2 for prompts. For experiment in Sec 4.3, we use sample 128 prompts, each with 16 trajectories. During training, we drop the KL loss term to improve the numerical stability. We use a learning rate of 1e-6, bfloat16 precision, FSDP backend for distributed training. We split the rollout data into 4 mini-batches for update. For the sake of numerical stability, we do pre-clip before computing the exponential of `log_prob` for a upperbound of 3.0.

For the main result in Fig 5, we test different rollout configurations with a `max_prompt_length` of 4096, training for 500 steps. We use 32 NVIDIA A800 GPUs, the longest training cost about 40 hours due to large scale validation per 10 steps.

For the ablation results in Fig 6, we use a tiny subset of MATH Level 3-5, training for 300 steps. Specifically, we sample 19 questions for every single type (133 instances in total). We use 8 NVIDIA A800 GPUs, the training cost about 30 hours

We test different rollout configurations:
(1) `max_num_turns=30`, `max_response_length=256`, `max_prompt_length=4096` (2) `max_num_turns=30`, `max_response_length=1024`, `max_prompt_length=3072`

And for the experiment of separate parameter in multi-turn ReMA, we iteratively train each agent with the same configuration as above, but with a switch interval of 10 steps, starting from the meta-thinking agent.

## E   Other Experiments

### E.1   Experiments of comparing to more baselines

To further comparing ReMA to other meta-thinking and multi-agent baselines, we focused on adding two representative baselines: Multi-Agent Debate (MAD) [Du et al., 2023] and Self-refine [Madaan et al., 2023].

For both two experiments, we use the prompts and implementation in Du et al. [2023]. Note that MAD and self-refine are inference-time methods, we test them under a computation close to ReMA. For MAD, we use 2 agents and let them interact 2 rounds. While for Self-Refine, we let the LLM to first propose an answer and then prompt it to critique and refine the answer only once. We use a temperature of 0 for most generations except the 1st round of MAD to maintain diversity. We summarize the additional baseline results in Table 3.

Self-Refine and MAD do not exhibit strong performance, which we attribute to insufficient scaling of test-time compute. However, under similar compute, single-turn ReMA enhances the system's problem-solving capabilities through RL training.

### E.2   Reward functions shape cross-agent behaviors

We also investigate the impact of different reward function designs on ReMA's behavior. In addition to the base reward setting described in Appendix C.2, we evaluate a consistency-based reward function using Qwen2.5-7B-Instruct. This reward function is designed to encourage the high-level agent to generate more detailed guidance. Indeed, we observe that the high-level agent trained in this manner produces more detailed solution steps compared to the one trained with the basic correctness format reward. However, we also find that this approach often leads to jailbreak behavior,

Table 3: Additional baselines reordered as **CoT**, **Self-Refine**, **MAD**, **ReMA (Ours)**. CoT comes from prior VRP results.

| Model | Benchmark | CoT | Self-Refine | MAD | ReMA (Ours) |
|---|---|---|---|---|---|
| **Llama3 -8B -Instruct** | MATH500 | 30.80 | 24.60 | 22.40 | 33.80 |
| | GSM8K | 67.48 | 65.58 | 66.76 | 79.38 |
| | AIME24 | 0.00 | 0.00 | 0.00 | 0.00 |
| | AMC23 | 2.50 | 2.50 | 11.25 | 22.50 |
| | Gaokao2023en | 22.34 | 17.40 | 20.00 | 28.57 |
| | Minerva Math | 8.82 | 12.13 | 11.76 | 13.97 |
| | Olympiad Bench | 8.44 | 4.59 | 4.67 | 8.89 |
| | Average | 20.05 | 18.12 | 19.55 | 26.73 |
| **Llama3.1 -8B -Instruct** | MATH500 | 50.80 | 26.60 | 39.20 | 53.20 |
| | GSM8K | 86.05 | 56.41 | 78.43 | 87.26 |
| | AIME24 | 10.00 | 0.00 | 5.00 | 13.33 |
| | AMC23 | 27.50 | 12.50 | 16.25 | 20.00 |
| | Gaokao2023en | 38.96 | 27.53 | 32.08 | 37.14 |
| | Minerva Math | 22.79 | 13.24 | 18.93 | 28.31 |
| | Olympiad Bench | 15.11 | 6.52 | 10.37 | 19.56 |
| | Average | 35.89 | 20.40 | 28.61 | 36.97 |
| **Qwen2.5 -7B -Instruct** | MATH500 | 75.00 | 76.40 | 72.60 | 74.40 |
| | GSM8K | 92.04 | 91.51 | 90.00 | 90.60 |
| | AIME24 | 6.67 | 6.67 | 11.67 | 20.00 |
| | AMC23 | 47.50 | 45.00 | 45.00 | 57.50 |
| | Gaokao2023en | 56.62 | 62.34 | 61.82 | 57.92 |
| | Minerva Math | 35.66 | 34.93 | 36.03 | 34.93 |
| | Olympiad Bench | 38.22 | 38.96 | 39.41 | 36.30 |
| | Average | 50.24 | 50.83 | 50.93 | 53.09 |

where the high-level agent tends to include the final answer within its output, compromising the intended hierarchical reasoning process.

Furthermore, we discover an interesting evolution of a pattern during training: although our experimental setup is designed for the high-level agent to provide a solution plan while the lower-level agent executes it, we find that under the consistency-based reward, the lower-level agent significantly increases its attempt of verification rather than straightforward execution. We observed a certain sentence commonly appearing in the low-level agent's responses: *"Let's go through the solution step by step to ensure clarity and correctness."* To quantify this effect, we track the frequency of it. We analyze this pattern across all mathematical test sets, sampling eight completions per question at a temperature of 0.7. Our empirical results have identified a **30x increase** of such self-verifying patterns in the model trained with the consistency-based reward compared to the one trained with the base reward. Moreover, we also observe additional variations of this pattern, e.g. *"Let's carefully re-evaluate the problem and solution to ensure accuracy and clarity."* These phrases indicate that the low-level agent is actively exploring to verify the detailed response provided by the high-level agent.

This suggests that (1) meta-thinking can not only emerge and be reinforced in the high-level agent but also in the low-level agent. During reinforcement learning (RL) training, the two agents develop **a novel problem-solving pattern** characterized by a **role reversal**. (2) **Consistency-based rewards promote a more self-corrective approach at the lower level**, potentially disrupting the intended separation of roles between planning and execution. For a detailed case study, refer to Appendix F.2.

### E.3 Detailed Training Curves on Different Datasets of Multi-turn ReMA

We show the detailed training curves of the multi-turn ReMA on different datasets in Fig. 8.

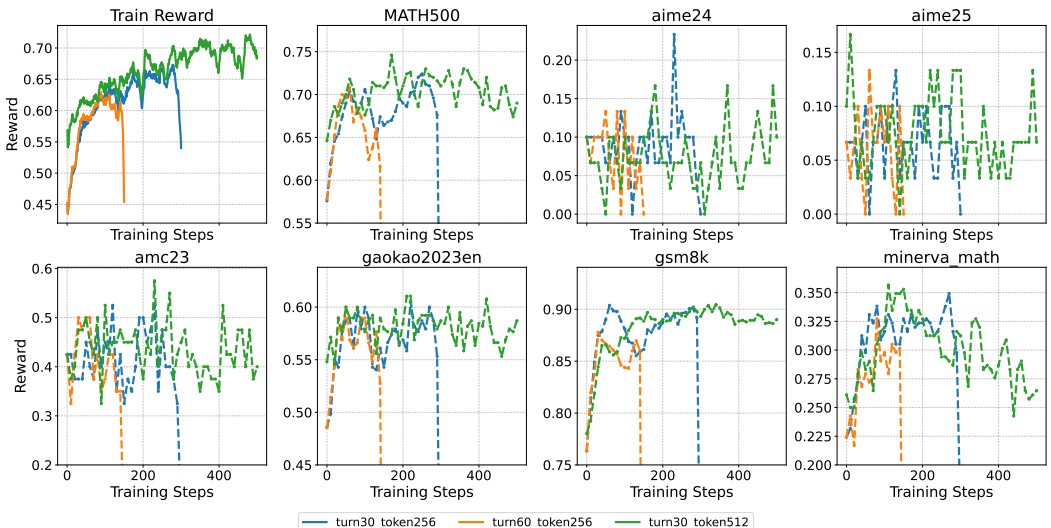

Figure 8: Detailed Training Curves on Different Datasets of Multi-turn ReMA

# F Qualitative Results

## F.1 High-level policy finds better plans

Here is an example of how a high-level policy alters the solving method of an LLM, increasing the likelihood of providing correct answers. As we can see from the following example, without the high-level policy, the LLM counts all integer coordinates, including those on the boundary, and then subtracts the boundary coordinates. In contrast, the high-level policy identifies a better approach, directly instructing the LLM to count only the coordinates strictly inside the boundary.

## F.2 Case study for Experiments in Section E.2

Fig. 10 and Fig. 11 show an case study of experiments in Sec. E.2.

Although both agents are prompted with the same instructions as in our main results, the consistency reward of the high-level agent significantly alters the learning dynamics. As illustrated in Fig. 10, the high-level agent generates detailed solution attempts rather than a strategic plan. Consequently, the low-level agent evolves to verify the high-level agent's solutions. This suggests that, during reinforcement learning (RL) training, the two agents develop **a novel problem-solving pattern** characterized by a **role reversal**, where one agent generates an answer while the other verifies it.

## F.3 Case study for Experiments in Section 4.2.2

Fig. 12 shows the detailed outputs of the two groups of LLMs trained in JSON action. The outputs finally converge to relative easier and shorter sentences, compared to our main experiments. This may due to insufficient capacity of a small LM to produce a valid JSON-format response while simultaneously exploring diverse reasoning strategies.

Consider the rectangular region with the following points as vertices: $(5, 4), (−5, 4), (−5, −4), (5, −4)$. How many points with integer coordinates will be strictly in the interior of this rectangular region?

**Output without high-level planning solves this question in a less optimal way, leading to a wrong answer:**

To determine how many points with integer coordinates will be strictly in the interior of the rectangular region with vertices at $(5, 4)$, $(−5, 4)$, $(−5, −4)$, and $(5, −4)$, we will follow these steps:
1. Identify the dimensions of the rectangle:
   - The width of the rectangle can be calculated as the distance between the x-coordinates of the vertices. The x-coordinates are $−5$ and $5$.

$$\text{Width} = 5 − (−5) = 10$$

   - The height of the rectangle can be calculated as the distance between the y-coordinates of the vertices. The y-coordinates are $4$ and $−4$.

$$\text{Height} = 4 − (−4) = 8$$

2. Calculate the total number of integer points: - The total number of integer points in the rectangle (including the boundary) can be calculated by considering the range of x and y values: - The x-values range from $−5$ to $5$, giving us 11 integer values $(−5, −4, −3, −2, −1, 0, 1, 2, 3, 4, 5)$. - The y-values range from $−4$ to $4$, giving us 9 integer values $(−4, −3, −2, −1, 0, 1, 2, 3, 4)$. - The total number of points in the rectangle (including the boundary) is:

$$\text{Total points} = 11 \times 9 = 99$$

3. Calculate the number of points on the boundary: - The boundary consists of the four sides of the rectangle. - The top and bottom sides (y = 4 and y = -4) each have 11 integer points (from $−5$ to 5). - The left and right sides (x = 5 and x = -5) each have 9 integer points (from $−4$ to 4). - Therefore, the total number of points on the boundary is:

$$\text{Boundary points} = 2 \times 11 + 2 \times 9 = 22 + 18 = 40$$

4. Calculate the number of strictly interior points: - To find the number of points strictly in the interior, we subtract the points on the boundary from the total number of points:

$$\text{Interior points} = \text{Total points} − \text{Boundary points} = 99 − 40 = 59$$

Thus, the number of points with integer coordinates that will be strictly in the interior of this rectangular region is $\boxed{59}$.

**Answer with high-level planning identifies the optimal sub-tasks for the LLM to solve, increasing the chance of answering the question correctly**

**High-level policy planning:** First, identify the boundaries of the rectangular region. Then, determine the integer coordinates that lie strictly within these boundaries. Finally, count the number of such points.

**Low-level policy reasoning following the high-policy plan:** To solve the problem, we will follow the steps outlined in the instructions.

Step 1: Identify the boundaries of the rectangular region. The vertices of the rectangular region are:- $(5, 4)$- $(−5, 4)$- $(−5, −4)$- $(5, −4)$ From these vertices, we can determine the boundaries:- The left boundary is $x = −5$.- The right boundary is $x = 5$.- The bottom boundary is $y = −4$.- The top boundary is $y = 4$.

Step 2: Determine the integer coordinates that lie strictly within these boundaries. For a point $(x, y)$ to be strictly inside the rectangle, it must satisfy:- $−5 < x < 5$- $−4 < y < 4$

Step 3: Count the integer coordinates that satisfy these conditions. Now, we will find the integer values for $x$ and $y$ that fall within these boundaries. For $x$ :- The integer values that satisfy $−5 < x < 5$ are: $−4, −3, −2, −1, 0, 1, 2, 3, 4$- This gives us 9 possible integer values. For $y$:- The integer values that satisfy $−4 < y < 4$ are: $−3, −2, −1, 0, 1, 2, 3$. This gives us 7 possible integer values.

Step 4: Calculate the total number of points. Now, we multiply the number of possible integer values for $x$ by the number of possible integer values for $y$:

$$\text{Total points} = 9 \times 7 = 63$$

Finally, the number of points with integer coordinates that lie strictly in the interior of the rectangular region is $\boxed{63}$.

Figure 9: Case Study comparing with and without high-level metacognition results.

# G  Prompts

## G.1  Single-turn ReMA prompts

### G.1.1  Prompts for JSON data collection

Prompt for metacognition reasoning to rewrite:

```
System prompt:
You are a math expert trying to solve mathematical problems.
Before answering a question, your task is to rewrite the original
question to make it clearer.
Provide your rewritten content in JSON format:
{{
"action":  "REWRITE", "output":  "{{clearer question content}}"
}}
Respond only with valid JSON. Do not write an introduction or
summary.
User prompt:
Here is the question:
[problem_text]
```

Prompt for metacognition reasoning to decompose:

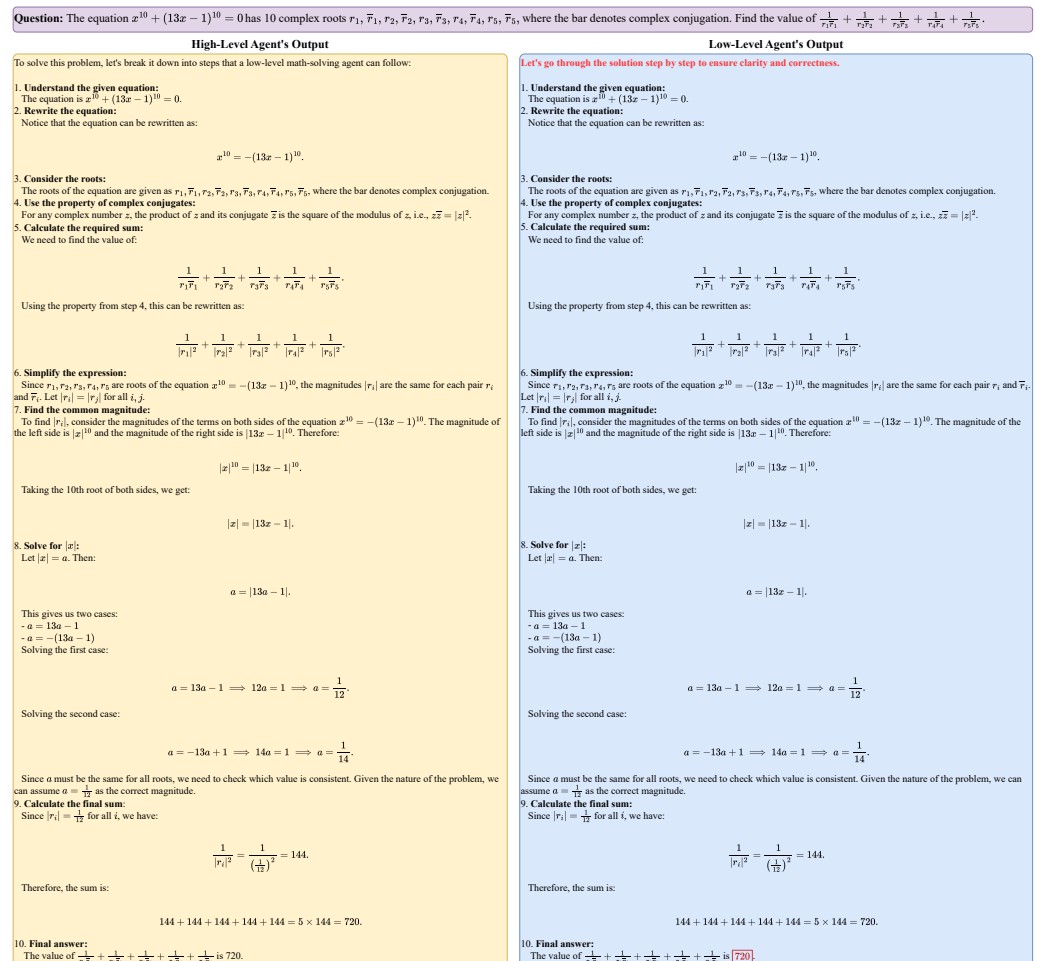

Figure 10: Case Study for consistency reward of high-level agent

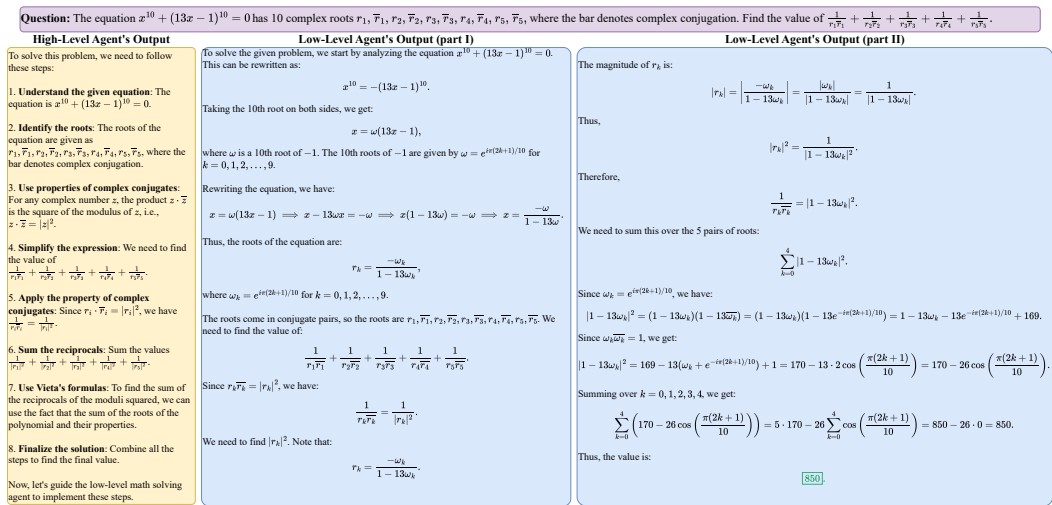

Figure 11: Case Study for base correctness reward of high-level agent

**Question:** The product of $3t^2 + 5t + a$ and $4t^2 + bt - 2$ is $12t^4 + 26t^3 - 8t^2 - 16t + 6$. What is $a + b$?

**Llama3.1-8B-Instruct training Output:**

```
{
    "action": "DECOMPOSE",
    "output": "Decompose the problem into finding the coefficients of the
product of two polynomials and then solving for a and b."
}
```

Step 1: To find the product of the two polynomials
$3t^2 + 5t + a$ and $4t^2 + bt - 2$,
we need to multiply each term of the first polynomial by each term of the
second polynomial and then combine like terms.
Step 2: The product of the two polynomials can be written as
$(3t^2 + 5t + a)(4t^2 + bt - 2)$.
Step 3: Using the distributive property, we multiply each term in the first
polynomial by each term in the second polynomial:
$(3t^2)(4t^2) + (3t^2)(bt) + (3t^2)(-2)$
$+(5t)(4t^2) + (5t)(bt) + (5t)(-2) + (a)(4t^2) + (a)(bt) + (a)(-2)$.
Step 4: Simplifying the products, we get
$12t^4 + 3bt^3 - 6t^2 + 20t^3 + 5bt^2 - 10t + 4at^2 + abt - 2a$.
Step 5: Combining like terms, we have
$12t^4 + (3b + 20)t^3 + (5b - 6 + 4a)t^2 + (ab - 10)t - 2a$.
Step 6: We are given that the product of the two polynomials is
$12t^4 + 26t^3 - 8t^2 - 16t + 6$.
We can equate the coefficients of the corresponding powers of $t$ in the two
expressions.
Step 7: Equating the coefficients of $t^3$, we get $3b + 20 = 26$.
Step 8: Equating the coefficients of $t^2$, we get $5b - 6 + 4a = -8$.
Step 9: Equating the constant terms, we get $-2a = 6$.
Step 10: Solving the equation $-2a = 6$, we find that $a = -3$.
Step 11: Substituting $a = -3$ into the equation $5b - 6 + 4a = -8$, we get
$5b - 6 + 4(-3) = -8$, which simplifies to $5b - 18 = -8$.
Step 12: Solving for $b$, we find that $5b = 10$, so $b = 2$.
Step 13: Now that we have found the values of $a$ and $b$, we can calculate
their sum: $a + b = -3 + 2 = -1$.

The final answer is: $\boxed{-1}$

**Llama3.2-1B-Instruct training Output:**

```
{
  "action": "EMPTY",
  "output": ""
}
```

Step 1: The problem asks us to find the values of $a$ and $b$ in the quadratic
expressions $3t^2 + 5t + a$ and $4t^2 + bt - 2$, respectively, given that their
product is $12t^4 + 26t^3 - 8t^2 - 16t + 6$.

Step 2: To solve this, we need to multiply the two quadratic expressions and
equate the resulting expression to the given product
$12t^4 + 26t^3 - 8t^2 - 16t + 6$.

Step 3: We start by multiplying the two quadratic expressions:
$(3t^2 + 5t + a)(4t^2 + bt - 2)$.

Step 4: This gives us
$12t^4 + (3b + 5)t^3 + (4a + 5b - 6)t^2 + (-2b + 10t + 2a)t - 2a$.

Step 5: We know that the coefficients of the resulting expression must be
equal to the coefficients of the given product $12t^4 + 26t^3 - 8t^2 - 16t + 6$.

Step 6: Equating the coefficients of $t^3$ gives us the equation $3b + 5 = 26$,
which simplifies to $3b = 21$ and thus $b = 7$.

Step 7: Equating the constant terms gives us $-2a = -6$, which simplifies
to $a = 3$.

Step 8: Now that we have the values of $b$ and $a$, we can find the sum $a + b$.

Step 9: Adding $a = 3$ and $b = 7$ gives us $a + b = 10$.

The final answer is: $\boxed{10}$

Figure 12: Case Study for interpretability experiments in Section 4.2.2

System prompt:
```
You are a math expert trying to solve mathematical problems.
Before answering a question, your task is to decompose the
original question to make it clearer.
Provide your rewritten content in JSON format:
{{
"action":  "DECOMPOSE", "output":  "{{decomposed question
content}}"
}}
Respond only with valid JSON. Do not write an introduction or
summary.
```
User prompt:
```
Here is the question:
[problem_text]
```

Prompt for generating final answers using on the question and metacognition reasoning:

System prompt:
```
You are a math expert tasked with solving problems step by step.
Follow the provided instructions precisely, showing all reasoning
and intermediate steps.
Present the final answer within \boxed{{}}.
```
User prompt:
```
Here is the question and instructions:
Question
[problem_text]
Provided Instruction
[instruction_text]
```

### G.1.2 Prompts for Math problems

**VRP prompt:**

System prompt:
You are a math expert tasked with solving problems step by step.
Present the final answer within \boxed{}.
User prompt:
Here is the question:
{Question}

**MRP prompt:**

System prompt:
You are a math expert tasked with solving problems.  When solving
a problem, your first task is to provide a high-level solution
plan as an instruction.  Then you need to follow the provided
instructions precisely, showing all reasoning and intermediate
steps.  Finally, you must present the final answer within
\boxed{}.
User prompt:
Here is the question:
{Question}

**MAMRP prompt:**
high-level agent:

System prompt:
You are a math expert specialized in solving mathematical
problems, you need to teach a weaker agent with minimal capability
in math how to solve a problem step-by-step.
Your task is to provide a high-level solution plan for the given
problem, in order to guide a low-level math solving agent to solve
the problem.
You can not directly answer the question.  You'll be punished if
you include any answer in your response.
You need to first think deeply in mind and output your final
instruction.
User prompt:
Here is the question:
{Question}

low-level agent:

System prompt:
You are a math expert tasked with solving problems step by step.
Follow the provided instructions precisely, showing all reasoning
and intermediate steps.  Present the final answer within \boxed{}.
User prompt:
Here is the question and instructions:

[Question]
{Question}
[End of Question]

[Provided Instruction]
{instruction}
[End of Instruction]

### G.1.3 Prompts for LLM-as-a-Judge problems

We adopt the prompts from Saha et al. [2025].

**VRP prompt:**

System prompt:
```
Please act as an impartial judge and evaluate the quality of the
responses provided by two AI assistants to the user question
displayed below.  You should choose the assistant that follows
the user's instructions and answers the user's question better.
Your evaluation should consider factors such as the helpfulness,
relevance, accuracy, depth, creativity, and level of detail of
their responses.  Begin your evaluation by comparing the two
responses and provide a short explanation.  Avoid any position
biases and ensure that the order in which the responses were
presented does not influence your decision.
Do not allow the length of the responses to influence your
evaluation.
Do not favor certain names of the assistants.  Be as objective
as possible.  After providing your explanation, output your final
verdict by strictly following this format:  "[[A]]" if assistant A
is better, "[[B]]" if assistant B is better.
```
User prompt:
```
[User Question]
{instruction}
[End of User Question]
[The Start of Assistant A's Answer]
{response_A}
[The End of Assistant A's Answer]
[The Start of Assistant B's Answer]
{response_B}
[The End of Assistant B's Answer]
```

**MRP prompt:**

System prompt:
Please act as an impartial judge and evaluate the quality of the
responses provided by two AI assistants to the user question
displayed below.  You should choose the assistant that follows
the user's instructions and answers the user's question better.
First of your task is to build an evaluation plan that can then
be executed to assess the response quality.  Whenever appropriate,
you can choose to also include a step-by-step reference answer as
part of the evaluation plan.
Enclose your evaluation plan between the tags "[Start of
Evaluation Plan]" and "[End of Evaluation Plan]".
After that, please act as an impartial judge and evaluate the
quality of the responses provided by two AI assistants to the user
question displayed below.  You should choose the assistant that
follows the user's instructions and answers the user's question
better.
Your evaluation should consider factors such as the helpfulness,
relevance, accuracy, depth, creativity, and level of detail of
their responses.  Begin your evaluation by comparing the two
responses and provide a short explanation.  Avoid any position
biases and ensure that the order in which the responses were
presented does not influence your decision.
Do not allow the length of the responses to influence your
evaluation.
Do not favor certain names of the assistants.  Be as objective
as possible.  After providing your explanation, output your final
verdict by strictly following this format:  "[[A]]" if assistant A
is better, "[[B]]" if assistant B is better.
User prompt:
[User Question]
{instruction}
[End of User Question]
[The Start of Assistant A's Answer]
{response_A}
[The End of Assistant A's Answer]
[The Start of Assistant B's Answer]
{response_B}
[The End of Assistant B's Answer]

**MAMRP prompt:** high-level agent:

System prompt:
We want to evaluate the quality of the responses provided by AI
assistants to the user question displayed below.
For that, your task is to help us build an evaluation plan that
can then be executed to assess the response quality.  Whenever
appropriate, you can choose to also include a step-by-step
reference answer as part of the evaluation plan.
User prompt:
[User Question]
{Question}
[End of User Question]

low-level agent:

System prompt:
Please act as an impartial judge and evaluate the quality of
the responses provided by two AI assistants to the user question
displayed below.  Your evaluation should be performed by following
the provided evaluation plan step-by-step.  Avoid copying the plan
when doing the evaluation.
Please also only stick to the given plan and provide explanation
of how the plan is executed to compare the two responses.
Avoid any position biases and ensure that the order in which the
responses were presented does not influence your decision.
Do not allow the length of the responses to influence your
evaluation.  Do not favor certain names of the assistants.  Be
as objective as possible.
After providing your evaluation, output your final verdict by
strictly following this format:  "[[A]]" if assistant A is better,
"[[B]]" if assistant B is better.
User prompt:
[User Question]
{instruction}
[End of User Question]
[The Start of Assistant A's Answer]
{response_A}
[The End of Assistant A's Answer]
[The Start of Assistant B's Answer]
{response_B}
[The End of Assistant B's Answer]
[The Start of Evaluation Plan]
{evaluation_plan}
[The End of Evaluation Plan]

## G.2 Multi-turn ReMA prompts

### G.2.1 SFT data collection of multi-turn MAMRP

System prompt:
You are classifying reasoning process data into two types of
thinking. You will be given a question-answer pair from a
reasoning dataset. Your task is to split all words into two
parts. These words are crucial for analyzing reasoning patterns,
so do not skip any details.
- **Meta-Thinking Agent (MTA):** Responsible for high-level
thought processes. This includes planning, evaluating steps,
expressing uncertainty, making observations, or setting goals.
Avoid detailed calculations. The content should be enclosed in
`<meta_thinking>` and `</meta_thinking>`.
- **Reasoning Agent (RA):** Responsible for detailed
problem-solving steps, such as calculations, logical deductions,
or breaking down a problem into subproblems. The content should
be enclosed in `<reasoning>` and `</reasoning>`.
**Rules to follow:**
1. **Do not assign large chunks of text to a single type of
thinking.** The reasoning process consists of small, nonlinear
thinking steps, so alternate appropriately between Meta-Thinking
and Reasoning steps.
2. **Keep the words from the original solution unmodified
whenever possible.** Words like "Wait," "Hmm," "But," etc.,
typically indicate Meta-Thinking and should be preserved.
3. **When finalizing the answer:**
- The **Meta-Thinking Agent (MTA)** must explicitly confirm the
answer before completion and output `[FINISH]`.
- The **Reasoning Agent (RA)** should then provide the final
answer in the correct format.
4. **Do not skip any reasoning steps, even if they seem
redundant, incorrect or irrelevant**
5. **Do not modify or remove any part of the original reasoning
process**, even if it seems redundant or repetitive. The goal is
to **preserve the exact flow of thought** as it naturally occurs.
6. **Retain all expressions such as "Wait," "Hmm," "But wait,"
etc., exactly as they appear. These indicate important cognitive
processes and should not be skipped or altered.**
Here are examples for you:
[Examples] ...
User prompt:
[Begin of Question]
{question}
[End of Question]
[Begin of Solution]
{solution}
[End of Solution]

### G.2.2 Prompt for math problems

**Meta-Thinking Agent (MTA):**

System prompt:
```
You are a meta-think agent that represents human high-level think
process, when solving a question, you will have a discussion with
human, each time you think about what to do next:  e.g.
- Exploring multiple angles and approaches
- Breaking down the solution into clear steps
- Continuously reflecting on intermediate results honestly and
adapt your strategy as you progress
- Backtracking when necessary
- Requesting exploration of multiple solutions individually
- Finally confirm the answer with the tag [FINISH]
```
User prompt:
```
{question}
```

**Reasoning Agent (RA):**

System prompt:
```
Please reason step by step follow the given instruction, when
asked to finalize your answer, put your answer within \boxed{}
```
User prompt:
```
{question}
{instruction}
```

