# OpenReview forum: "ReMA: Learning to Meta-Think for LLMs with Multi-agent Reinforcement Learning"
_NeurIPS.cc/2025/Conference — NeurIPS 2025 poster_

### Official Review · Reviewer_5jA5 · 2025-07-02

**Clarity:** 3
**Significance:** 3
**Originality:** 3
**Rating:** 5
**Confidence:** 2

**Summary:**

The use ideas from MARL to induce improvements in meta-thinking for LLMs.  They show results indicating that this approach is better than standard meta-thinking approaches (e.g. CoT).

**Questions:**

My main concern is that the authors haven't compared their method to a spectrum of state-of-the-art meta-thinking methods in the literature.  I am not enough of an expert in the domain to know what the perfectly right comparisons are, but many such methods have been proposed.     Some potential candidates are:

-- Tree of Thought
-- self-consistency - you run the prompt a bunch of times, let the model make different chains, then majority-vote the answers. (wang et al., 2022, https://arxiv.org/abs/2203.11171)
-- least-to-most prompting - break the hard question into sub-questions, solve them in order, add the pieces together at the end. (zhou et al., 2022, https://arxiv.org/abs/2205.10625)
-- self-refine - model drafts an answer, critiques itself, rewrites, repeat. (madaan et al., 2023, https://arxiv.org/abs/2303.17651)
-- reflexion - llm writes little “what i learned” memos after each run and reads them next time, improving without training. (shinn et al., 2023, https://arxiv.org/abs/2303.11366)

I would like to either see the authors compare to some of those methods, or some others of their choosing, or explain very clearly why they don't have to.

**Ethical Concerns:**

["NO or VERY MINOR ethics concerns only"]

**Final Justification:**

The authors addressed some of my concerns in the rebuttal/discussion process, so I am raising my score.

**Limitations:**

Seems ok to me -- with the exception of the limitations of their method as compared to alternatives (mentioned above in my comments).

**Quality:**

3

**Strengths And Weaknesses:**

Strengths:

-- The idea of the paper is clear and interesting
-- the benchmarks are meaningful
-- the ablations are pretty comprehensive
-- the results are compelling in relation to the shown baselines.

Weaknesses
-- I believe there are tons of other variants of meta-rethinking in the literature besides CoT.   It seems to me (though I am not really a very very expert person in this domain) that it would be great to compare to some of these other methods.   To me this seems like a very important point.

---

> ### Author Rebuttal · Authors · 2025-07-31
>
> ### **Weakness 1: Not compared to meta-thinking baselines**
>
> We sincerely thank the reviewers for their thoughtful and constructive comments. We have carefully addressed the requests regarding additional baseline experiments. Due to time and computational constraints, we focused on adding two representative baselines: **Multi-Agent Debate (MAD)** and **Self-refine**, one as a strong multi-agent meta-reasoning baseline and the other as a strong single-agent meta-thinking baseline.
>
> For both two experiments, we use the prompts and implementation in [1]. Note that MAD and self-refine are inference-time methods, we test them under a computation close to ReMA. For MAD, we use 2 agents and let them interact 2 rounds. While for Self-Refine, we let the LLM to first propose an answer and then prompt it to critique and refine the answer only once. We use a temperature of 0 for most generations except the 1st round of MAD to maintain diversity.
>
> We summarize the additional baseline results below:
>
> ##### Multi-agent Debat (MAD) vs Self-refine vs ReMA (Ours)
>
> | Benchmark       | Qwen2.5-7B-Instruct         |               |               | LLaMA3.1-8B-Instruct      |               |               | LLaMA3-8B-Instruct        |               |               |
> |-----------------|-----------------------------|---------------|-----------------|----------------------------|---------------|-----------------|----------------------------|---------------|-----------------|
> |                 | **ReMA**                    | **MAD**       | **Self-refine** | **ReMA**                   | **MAD**       | **Self-refine** | **ReMA**                   | **MAD**       | **Self-refine** |
> | MATH500         | 74.40%                       | 72.60%        | **76.40%**      | **53.20%**                  | 39.20%        | 26.60%          | **33.80%**                  | 22.40%        | 24.60%          |
> | GSM8K           | 90.60%                       | 90.00%        | **91.51%**      | **87.26%**                  | 78.43%        | 56.41%          | **79.38%**                  | 66.76%        | 65.58%          |
> | AIME24          | **20.00%**                   | 11.67%        | 6.67%           | **13.33%**                  | 5.00%         | 0.00%           | **0.00%**                   | **0.00%**     | **0.00%**       |
> | AMC23           | **57.50%**                   | 45.00%        | 45.00%          | **20.00%**                  | 16.25%        | 12.50%          | **22.50%**                  | 11.25%        | 2.50%           |
> | Gaokao2023en    | 57.92%                       | 61.82%        | **62.34%**      | **37.14%**                  | 32.08%        | 27.53%          | **28.57%**                  | 20.00%        | 17.40%          |
> | Minerva_math    | 34.93%                       | **36.03%**    | 34.93%          | **28.31%**                  | 18.93%        | 13.24%          | **13.97%**                  | 11.76%        | 12.13%          |
> | OlympiadBench   | 36.30%                       | **39.41%**    | 38.96%          | **19.56%**                  | 10.37%        | 6.52%           | **8.89%**                   | 4.67%         | 4.59%           |
> | **AVG**         | **53.09%**                   | 50.93%        | 50.83%          | **36.97%**                  | 28.61%        | 20.40%          | **26.73%**                  | 19.55%        | 18.12%          |
>
>
>
> Compared to these inference-time methods, our proposed method (**ReMA**) achieves overall higher performance. This clearly demonstrates the advantage of our multi-agent reinforcement learning-based meta-thinking training strategy.
>
> While MAD represents a strong multi-agent reasoning baseline, and self-refine serves as a classic single-agent meta-thinking method, our method still outperforms both significantly. Specifically, **ReMA** leverages structured multi-agent interaction optimized by reinforcement learning, enabling it to systematically improve reasoning capabilities beyond inference-only methods.
>
> **Reference**
>
> [1] Du et al., "Improving factuality and reasoning in language models through multi-agent debate," ICML2023.

---

> ### Comment · Reviewer_5jA5 · 2025-08-03
> **score raise**
>
> I would like to raise my score to "accept" -- based on the rebuttal (which I discussed elsewhere in a different comment).

---

> > ### Author Response · Authors · 2025-08-04
> >
> > Thank you very much for your careful evaluation and your insightful suggestions. We truly appreciate your support and consideration.

---

### Official Review · Reviewer_uS42 · 2025-07-02

**Clarity:** 3
**Significance:** 3
**Originality:** 2
**Rating:** 3
**Confidence:** 3

**Summary:**

This paper introduces the ReMA framework, which uses Multi-agent Reinforcement Learning (MARL) to encourage meta-thinking in Large Language Models (LLMs). Meta-thinking allows models to not only perform reasoning but also to evaluate and control their reasoning processes. The paper proposes a two-agent system where a high-level agent handles meta-thinking (strategic oversight) and a low-level agent handles detailed execution.

**Questions:**

See weakness

**Ethical Concerns:**

["NO or VERY MINOR ethics concerns only"]

**Limitations:**

See weakness

**Quality:**

3

**Strengths And Weaknesses:**

Strengths:

The concept of separating meta-thinking from reasoning and executing it through two distinct agents is a novel approach. This allows for more efficient exploration and training compared to conventional methods, where a single agent is responsible for both tasks.

The paper presents thorough experiments in both single-turn and multi-turn scenarios, achieving impressive results across various benchmarks. This strongly supports the effectiveness of the proposed approach.

The framework demonstrates potential for tackling more complex, long-term reasoning tasks, thanks to its capability to function in both single-turn and multi-turn interaction settings.

Weaknesses：

The application of MARL in multi-agent environments can be computationally intensive, and scalability may become an issue as the number of agents or the complexity of tasks grows. This challenge needs to be addressed to ensure the approach remains effective in larger-scale scenarios.

While the paper compares ReMA with single-agent reinforcement learning and other baseline methods, it would be beneficial to include comparisons with existing multi-agent meta-reasoning approaches to better highlight the unique advantages of ReMA.

The paper mentions that the multi-agent approach improves exploration; could the authors provide more details on how exploration is structured in the multi-agent setting compared to single-agent models?

Are there any plans to extend ReMA for multi-agent cooperation in collaborative tasks, such as cooperative problem-solving or resource-sharing scenarios?

---

> ### Author Rebuttal · Authors · 2025-07-31
>
> ### **Weakness 1: Concern about computational efficiency**
>
> We appreciate the reviewer’s insightful comment about computational efficiency and scalability. We fully acknowledge this limitation. This has always been one of the primary challenges in MARL, and yet mainstream training frameworks such as VeRL still offer limited native support for multi‑agent RL.
> However, there are already increasing efforts trying to solve this issue: e.g.
> (i) parameter sharing across agents (we tried in this work)
> (ii) PD disaggregation that reuses cached states, and
> (iii) an asynchronous actor–learner pipeline.
> These design choices allow our method to keep near‑linear training speed as the number of agents or task complexity grows. In fact, the community is likewise beginning to focus on scalable solutions, such as Multiverse[1] and SGLang[2]. And we are willing to further iterate on our codebase moving forward. We hope these additions resolve the reviewer’s concern and are grateful for the opportunity to clarify.
>
>
> ### **Weakness 2: Comparisons with existing multi-agent meta-reasoning approaches**
>
> We sincerely thank the reviewers for their thoughtful and constructive comments. We have carefully addressed the requests regarding additional baseline experiments. Due to time and computational constraints, we focused on adding two representative baselines: **Multi-Agent Debate (MAD)** and **Self-refine**, to cover key concerns raised by the reviewers.
>
> For both two experiments, we use the prompts and implementation in [3]. Note that MAD and self-refine are inference-time methods, we test them under a computation close to ReMA. For MAD, we use 2 agents and let them interact 2 rounds. While for Self-Refine, we let the LLM to first propose an answer and then prompt it to critique and refine the answer only once. We use a temperature of 0 for most generations except the 1st round of MAD to maintain diversity.
>
> We summarize the additional baseline results below:
>
> ##### Multi-agent Debat (MAD) vs Self-refine vs ReMA (Ours)
>
> | Benchmark       | Qwen2.5-7B-Instruct         |               |               | LLaMA3.1-8B-Instruct      |               |               | LLaMA3-8B-Instruct        |               |               |
> |-----------------|-----------------------------|---------------|-----------------|----------------------------|---------------|-----------------|----------------------------|---------------|-----------------|
> |                 | **ReMA**                    | **MAD**       | **Self-refine** | **ReMA**                   | **MAD**       | **Self-refine** | **ReMA**                   | **MAD**       | **Self-refine** |
> | MATH500         | 74.40%                       | 72.60%        | **76.40%**      | **53.20%**                  | 39.20%        | 26.60%          | **33.80%**                  | 22.40%        | 24.60%          |
> | GSM8K           | 90.60%                       | 90.00%        | **91.51%**      | **87.26%**                  | 78.43%        | 56.41%          | **79.38%**                  | 66.76%        | 65.58%          |
> | AIME24          | **20.00%**                   | 11.67%        | 6.67%           | **13.33%**                  | 5.00%         | 0.00%           | **0.00%**                   | **0.00%**     | **0.00%**       |
> | AMC23           | **57.50%**                   | 45.00%        | 45.00%          | **20.00%**                  | 16.25%        | 12.50%          | **22.50%**                  | 11.25%        | 2.50%           |
> | Gaokao2023en    | 57.92%                       | 61.82%        | **62.34%**      | **37.14%**                  | 32.08%        | 27.53%          | **28.57%**                  | 20.00%        | 17.40%          |
> | Minerva_math    | 34.93%                       | **36.03%**    | 34.93%          | **28.31%**                  | 18.93%        | 13.24%          | **13.97%**                  | 11.76%        | 12.13%          |
> | OlympiadBench   | 36.30%                       | **39.41%**    | 38.96%          | **19.56%**                  | 10.37%        | 6.52%           | **8.89%**                   | 4.67%         | 4.59%           |
> | **AVG**         | **53.09%**                   | 50.93%        | 50.83%          | **36.97%**                  | 28.61%        | 20.40%          | **26.73%**                  | 19.55%        | 18.12%          |
>
>
>
> Compared to these inference-time methods, our proposed method (**ReMA**) achieves overall higher performance. This clearly demonstrates the advantage of our multi-agent reinforcement learning-based meta-thinking training strategy.
>
> While MAD represents a strong multi-agent reasoning baseline, and self-refine serves as a classic single-agent meta-thinking method, our method still outperforms both significantly. Specifically, **ReMA** leverages structured multi-agent interaction optimized by reinforcement learning, enabling it to systematically improve reasoning capabilities beyond inference-only methods.
>
>
>
> ### **Question 1: More details on how exploration is structured in the multi-agent setting compared to single-agent models?**
>
> Thank you for your helpful and constructive suggestions! Below we outline the two main mechanisms through which ReMA structures exploration more efficiently.
> (i) Hierarchical decomposition shrinks the search space: single‑agent RL must discover high‑level planning, sub‑goal tracking, reflection and correction within one long autoregressive rollout. The agent therefore explores an undifferentiated action space that mixes meta decisions with low‑level reasoning tokens, leading to sparse rewards and early convergence to sub‑optimal patterns; ReMA decouples those roles. A high‑level meta‑thinking agent proposes a compact plan, while a low‑level reasoning agent executes it. Because each agent now solves a simpler sub‑problem, the effective search space factorises; this yields more diverse trajectories and faster policy improvement, as shown in Figure 3.
> (ii) Context‑engineered workflow accelerates computation & execution: In multi-agent systems, the agents can exchange succinct messages instead of the entire history. Consequently the context window processed per forward pass is shorter, which saves the computations. Moreover, multiple agents can solve sub-tasks in parallel, which accelerates the task execution process.
>
> ### **Question 2: Are there any plans to extend ReMA for multi-agent cooperation in collaborative tasks, such as cooperative problem-solving or resource-sharing scenarios?**
>
> Thanks for your spot-on insight. We are actively exploring broader applications of the ReMA framework, and a promising next step is an AI‑Scientist‑style expert ensemble, in which ReMA’s meta‑agent is promoted to a principal investigator who assigns sub‑problems to a team of domain‑expert agents with different tools working cooperatively.
>
>
> **Reference**
>
> [1] Yang et al., "Multiverse: Your Language Models Secretly Decide How to Parallelize and Merge Generation," arXiv:2506.09991v2, 2025.
>
> [2] Zheng et al., "SGLang: Efficient Execution of Structured Language Model Programs," arXiv:2312.07104v2, 2023.
>
> [3] Du et al., "Improving factuality and reasoning in language models through multi-agent debate," ICML2023.

---

> > ### Comment · Reviewer_5jA5 · 2025-08-03
> > **satisfied**
> >
> > This is a very helpful response.   I think especially the concern in weakness 2 was addressed pretty effectively. I am raising my score.

---

> > > ### Author Response · Authors · 2025-08-04
> > >
> > > We sincerely appreciate your positive feedback and acknowledgment of our efforts to address your concerns. Thank you for your thoughtful comments that have significantly improved our paper.

---

> > ### Comment · Reviewer_uS42 · 2025-08-04
> > **Comments after Author Discussion**
> >
> > My concerns are well addressed. Therefore I will recommend this work for accepting.

---

### Official Review · Reviewer_ydZo · 2025-07-03

**Clarity:** 3
**Significance:** 3
**Originality:** 3
**Rating:** 5
**Confidence:** 3

**Summary:**

This paper introduces Reinforced Meta-thinking Agents (ReMA), a novel framework that incorporates Multi-Agent Reinforcement Learning to enhance the reasoning capabilities of LLMs by fostering meta-thinking—the ability to monitor, evaluate, and control reasoning processes. Specifically, ReMA separates the reasoning process into two specialized agents: a high-level meta-thinking agent that provides strategic oversight and plans, and a low-level reasoning agent that executes detailed problem-solving steps.

By aligning their objectives through reinforcement learning, ReMA outperforms single-agent baselines on complex reasoning tasks like mathematical benchmarks. The framework also extends to multi-turn interactions, leveraging parameter sharing and turn-level optimization for improved efficiency. Extensive experiments and ablation studies demonstrate the effectiveness of ReMA in boosting reasoning accuracy, adaptability, and collaboration between agents, providing new insights into structured multi-agent reasoning for LLMs.

**Questions:**

See above

**Ethical Concerns:**

["NO or VERY MINOR ethics concerns only"]

**Final Justification:**

My concerns were adequately addressed in the rebuttal and discussion.

**Limitations:**

yes

**Quality:**

3

**Strengths And Weaknesses:**

**Strengths:**

1. The paper introduces an innovative approach to training large language models (LLMs) by explicitly decoupling strategic oversight (meta-thinking) from problem-solving (reasoning).
2. The proposed method achieves state-of-the-art performance on challenging reasoning benchmarks, including out-of-distribution (OOD) datasets such as GSM8K and AIME24.
3. The experimental section provides comprehensive theoretical support and includes ablation studies to dissect the dynamics and contributions of the high-level (strategic) and low-level (problem-solving) agents.

**Weakness:**

1. Missing important baselines: The paper does not evaluate the proposed method using baselines where ReMA is applied while keeping the MAMRP rollout fixed and freezing the weights of one agent. This additional experiment could further validate the impact of Leader-Follower game interactions and strengthen empirical claims.

---

> ### Author Rebuttal · Authors · 2025-07-31
>
> ### **Weakness 1: Missing important baselines**
>
> We sincerely thank the reviewers for their thoughtful and constructive comments. We have carefully addressed the requests regarding additional baseline experiments. Due to time and computational constraints, we focused on adding two representative baselines: **Multi-Agent Debate (MAD)** and **Self-refine**, to cover key concerns raised by the reviewers.
>
> For both two experiments, we use the prompts and implementation in [1]. Note that MAD and self-refine are inference-time methods, we test them under a computation close to ReMA. For MAD, we use 2 agents and let them interact 2 rounds. While for Self-Refine, we let the LLM to first propose an answer and then prompt it to critique and refine the answer only once. We use a temperature of 0 for most generations except the 1st round of MAD to maintain diversity.
>
> We summarize the additional baseline results below:
>
> ##### Multi-agent Debat (MAD) vs Self-refine vs ReMA (Ours)
>
> | Benchmark       | Qwen2.5-7B-Instruct         |               |               | LLaMA3.1-8B-Instruct      |               |               | LLaMA3-8B-Instruct        |               |               |
> |-----------------|-----------------------------|---------------|-----------------|----------------------------|---------------|-----------------|----------------------------|---------------|-----------------|
> |                 | **ReMA**                    | **MAD**       | **Self-refine** | **ReMA**                   | **MAD**       | **Self-refine** | **ReMA**                   | **MAD**       | **Self-refine** |
> | MATH500         | 74.40%                       | 72.60%        | **76.40%**      | **53.20%**                  | 39.20%        | 26.60%          | **33.80%**                  | 22.40%        | 24.60%          |
> | GSM8K           | 90.60%                       | 90.00%        | **91.51%**      | **87.26%**                  | 78.43%        | 56.41%          | **79.38%**                  | 66.76%        | 65.58%          |
> | AIME24          | **20.00%**                   | 11.67%        | 6.67%           | **13.33%**                  | 5.00%         | 0.00%           | **0.00%**                   | **0.00%**     | **0.00%**       |
> | AMC23           | **57.50%**                   | 45.00%        | 45.00%          | **20.00%**                  | 16.25%        | 12.50%          | **22.50%**                  | 11.25%        | 2.50%           |
> | Gaokao2023en    | 57.92%                       | 61.82%        | **62.34%**      | **37.14%**                  | 32.08%        | 27.53%          | **28.57%**                  | 20.00%        | 17.40%          |
> | Minerva_math    | 34.93%                       | **36.03%**    | 34.93%          | **28.31%**                  | 18.93%        | 13.24%          | **13.97%**                  | 11.76%        | 12.13%          |
> | OlympiadBench   | 36.30%                       | **39.41%**    | 38.96%          | **19.56%**                  | 10.37%        | 6.52%           | **8.89%**                   | 4.67%         | 4.59%           |
> | **AVG**         | **53.09%**                   | 50.93%        | 50.83%          | **36.97%**                  | 28.61%        | 20.40%          | **26.73%**                  | 19.55%        | 18.12%          |
>
>
>
> Compared to these inference-time methods, our proposed method (**ReMA**) achieves overall higher performance. This clearly demonstrates the advantage of our multi-agent reinforcement learning-based meta-thinking training strategy.
>
> While MAD represents a strong multi-agent reasoning baseline, and self-refine serves as a classic single-agent meta-thinking method, our method still outperforms both significantly. Specifically, **ReMA** leverages structured multi-agent interaction optimized by reinforcement learning, enabling it to systematically improve reasoning capabilities beyond inference-only methods.
>
> **Reference**
>
> [1] Du et al., "Improving factuality and reasoning in language models through multiagent debate," ICML2023.

---

> > ### Comment · Reviewer_ydZo · 2025-08-03
> > **Comments after Author Discussion**
> >
> > Thank you for your response. I don't have further questions.

---

> > > ### Author Response · Authors · 2025-08-04
> > >
> > > Thank you very much for your time and your consideration of our responses. We appreciate your efforts and constructive feedback throughout the review process.

---

### Official Review · Reviewer_2Abh · 2025-07-06

**Clarity:** 4
**Significance:** 3
**Originality:** 4
**Rating:** 3
**Confidence:** 4

**Summary:**

The paper treats meta thinking for LLMs as a multi-agent learning problem that can be trained alongside the standard generation process with RL and leads to strong results on benchmarks.

**Questions:**

1. Could the authors suggest a method of incorporating intermediate rewards/learnt values to fix the credit assignment issue and try out small scale experiments for this?

2. How does ReMA compare to simply prompting a single model with meta-thinking instructions and then performing RL on the subsequent model to best use these steps?

3. Could the authors try replacing the meta-thinking agent with some other baseline agent as an ablation? Is it really the meta-thinking that is helpful in increasing performance or spending more inference compute?

I would be happy to increase my score once these questions are answered.

**Ethical Concerns:**

["NO or VERY MINOR ethics concerns only"]

**Final Justification:**

I have addressed the rebuttals in my comment below and would like to note that they are not sufficient. I recommend rejecting the paper due to the issues highlighted in my comments.

**Limitations:**

Yes

**Quality:**

3

**Strengths And Weaknesses:**

Strengths

1. The idea of focusing on a meta-thinking agent in a multi-agent system and training end-to-end is novel and interesting.

2. The method achieves good results on a set of models and benchmarks.

3. The turn-level ratio clipping seems like a useful solution to prevent degenerate behaviors/collapses by treating each conversational turn as one action (not necessarily novel but an important solution in the method).

Weaknesses
1. Credit assignment remains an issue. A good meta-thought but poor reasoning steps would lead to negative rewards for both. This severely limits training efficiency. The state drift because of this is an important concern. It would be interesting to see a credit assignment technique like (https://arxiv.org/abs/2410.08115 or https://arxiv.org/pdf/2412.01928) used.

2. The improvements are not very high and not necessarily statistically significant. Standard RL baselines on Qwen models have often led to higher results on benchmarks than reported here.

3. The method seems quite fragile to different configurations and hyperparameters.

---

> ### Author Rebuttal · Authors · 2025-07-31
>
> ### **Weakness 1: Credit assignment remains an issue & Question 1: Suggestion of a method of incorporating intermediate rewards/values**
>
> Thank you for your constructive review and highlighting the credit‐assignment challenge.
> Here we address both Weakness 1 credit assignment issue and Question 1 incorporating intermediate rewards/values.
>
> As you point out, “poor reasoning steps would lead to negative rewards for both.” In our manuscript, we emphasize that we develop two agents collaboratively, where the meta‐thinking agent generates guidance for the reasoning agent. When the reward is negative, the agents search for a better collaborative strategy.
>
> For the papers you recommand (https://arxiv.org/pdf/2410.08115 and https://arxiv.org/pdf/2412.01928), they estimate the value of each turn using a Monte Carlo method, which we agree is very promising and plan to explore in future work. For example, sampling multiple reasoning trajectories for each meta‑thinking decision and averaging their rewards to compute the advantage, yielding more accurate credit assignment. We will discuss these works in our manuscripts.
>
> We also propose two additional methods for intermediate rewards/values.
> 1. Baseline reuse: We can leverage our existing “no‑meta‑thinking” samples as a baseline to reduce reward variance and support credit assignment.
> 2. Tabular Markov‑chain modeling: For tasks with a limited state space, we can treat the problem as a tabular Markov chain, akin to GiGPO[1] , and compute more accurate state values for advantage estimation.
>
> ### **Weakness 2: Improvement is not significant. RL baselines on Qwen models have often have higher results.**
>
> We thank the reviewer for their thoughtful comments. As shown in Section 4.2 Table 1, ReMA achieves substantial improvements on individual hard benchmarks, notably: AMC23 with Llama3-8B-Instruct: ReMA → +20.00% (2.50% → 22.50%), AIME24 with Qwen2.5-7B-Instruct: ReMA → +13.33% (6.67% → 20.00%). These results demonstrate large gains on challenging tasks. Our ablation study in Section 4.3.1 (Fig 6) further disentangles the contributions of turn‑level ratio clipping and the learned meta‑thinking policy, confirming that the observed improvement is directly attributable to the guidance provided by the meta‑thinking agent.
>
> We acknowledge that Qwen models fine‑tuned with RL can achieve larger improvements. However, most recent work applies RL to base models, which start from a lower initial performance and are hard to sample MAMRP-style data for training. In contrast, our approach leverages the full potential of already capable models and further enhances their reasoning ability.
>
> ### **Weakness 3: The method seems quite fragile to different configurations and hyperparameters.**
>
> Thank you for highlighting these important points. We agree that multi‑turn ReMA can be sensitive to some hyper-parameters and training configurations. That’s why we introduced turn‑level ratio clipping combined with parameter sharing between agents, which noticeably reduces collapse and stabilizes learning (Section 4.3.1, Question 5 and Figure 6). We are actively developing more robust credit‑assignment mechanisms and training recipes to further enhance stability.
>
> ### **Question 1: Suggestion of a method of incorporating intermediate rewards/values**
>
> Thanks for your constructive comments. Please see above reply to Weakness 1 and Question 1.
>
> ### **Question 2: How does ReMA compare to simply prompting a single model with meta-thinking instructions and then performing RL on the subsequent model to best use these steps?**
>
> We appreciate the reviewer’s detailed critique. We view the question from two perspectives: (1) the high‑level meta‑thinking agent, specifically, the quality of the meta‑thinking instructions it generates; and (2) the low‑level reasoning agent, its ability to follow those instructions and the instructions’ suitability for that agent.
>
> In our manuscript, we address these two aspects in separate sections respectively. In Section 4.2.1 and Figure 3, especially the comparison between the first and third settings, `RL from base` and `RL under Meta-thinking`. Figure 3 shows that this meta‑thinking guidance significantly boosts performance, demonstrating that the good instructions help with RL training of the reasoning agent. In Section 4.2.2 and Figure 4, we restrict the meta‑thinking instructions to “DECOMPOSE”, “REWRITE”, or “EMPTY”. Different models eventually converge to different meta-thinking instructions, the small model selects “EMPTY”, and the large model adapts to problem difficulties at the end, showing that we need different meta-thinking instructions for different models. We hope these experiments fully address your concerns.
>
> To eliminate confusion, we will add at the end of Section 4.2.2: “Sections 4.2.1 and 4.2.2 together demonstrate that RL with high-quality meta-thinking instructions enhances performance, and that the model subsequently selects instructions appropriate to its own capability.”
>
> ### **Question 3: Replacing Meta-thinking Agent with another baseline agent**
>
> We sincerely thank the reviewers for their thoughtful and constructive comments. We have carefully addressed the requests regarding additional baseline experiments. Due to time and computational constraints, we focused on adding two representative baselines: **Multi-Agent Debate (MAD)** and **Self-refine**, to cover key concerns raised by the reviewers.
>
> For both two experiments, we use the prompts and implementation in [2]. Note that MAD and self-refine are inference-time methods, we test them under a computation close to ReMA. For MAD, we use 2 agents and let them interact 2 rounds. While for Self-Refine, we let the LLM to first propose an answer and then prompt it to critique and refine the answer only once. We use a temperature of 0 for most generations except the 1st round of MAD to maintain diversity.
>
> We summarize the additional baseline results below:
>
> ##### Multi-agent Debat (MAD) vs Self-refine vs ReMA (Ours)
>
> | Benchmark       | Qwen2.5-7B-Instruct         |               |               | LLaMA3.1-8B-Instruct      |               |               | LLaMA3-8B-Instruct        |               |               |
> |-----------------|-----------------------------|---------------|-----------------|----------------------------|---------------|-----------------|----------------------------|---------------|-----------------|
> |                 | **ReMA**                    | **MAD**       | **Self-refine** | **ReMA**                   | **MAD**       | **Self-refine** | **ReMA**                   | **MAD**       | **Self-refine** |
> | MATH500         | 74.40%                       | 72.60%        | **76.40%**      | **53.20%**                  | 39.20%        | 26.60%          | **33.80%**                  | 22.40%        | 24.60%          |
> | GSM8K           | 90.60%                       | 90.00%        | **91.51%**      | **87.26%**                  | 78.43%        | 56.41%          | **79.38%**                  | 66.76%        | 65.58%          |
> | AIME24          | **20.00%**                   | 11.67%        | 6.67%           | **13.33%**                  | 5.00%         | 0.00%           | **0.00%**                   | **0.00%**     | **0.00%**       |
> | AMC23           | **57.50%**                   | 45.00%        | 45.00%          | **20.00%**                  | 16.25%        | 12.50%          | **22.50%**                  | 11.25%        | 2.50%           |
> | Gaokao2023en    | 57.92%                       | 61.82%        | **62.34%**      | **37.14%**                  | 32.08%        | 27.53%          | **28.57%**                  | 20.00%        | 17.40%          |
> | Minerva_math    | 34.93%                       | **36.03%**    | 34.93%          | **28.31%**                  | 18.93%        | 13.24%          | **13.97%**                  | 11.76%        | 12.13%          |
> | OlympiadBench   | 36.30%                       | **39.41%**    | 38.96%          | **19.56%**                  | 10.37%        | 6.52%           | **8.89%**                   | 4.67%         | 4.59%           |
> | **AVG**         | **53.09%**                   | 50.93%        | 50.83%          | **36.97%**                  | 28.61%        | 20.40%          | **26.73%**                  | 19.55%        | 18.12%          |
>
>
>
> Compared to these inference-time methods, our proposed method (**ReMA**) achieves overall higher performance. This clearly demonstrates the advantage of our multi-agent reinforcement learning-based meta-thinking training strategy.
>
> While MAD represents a strong multi-agent reasoning baseline, and self-refine serves as a classic single-agent meta-thinking method, our method still outperforms both significantly. Specifically, **ReMA** leverages structured multi-agent interaction optimized by reinforcement learning, enabling it to systematically improve reasoning capabilities beyond inference-only methods.
>
> **Reference**
>
> [1] Feng et al., "Group-in-group policy optimization for llm agent training," arXiv:2505.10978, 2025.
>
> [2] Du et al., "Improving factuality and reasoning in language models through multi-agent debate," ICML2023.

---

> > ### Author Response · Authors · 2025-08-05
> >
> > Dear reviewer 2Abh, we would like to gently remind you to share your response to the rebuttal and complete the mandatory acknowledgement when convenient. If there are any remaining concerns, we’d be grateful for the chance to address them before the review deadline. Thank you again for your time and thoughtful feedback.

---

> > ### Comment · Reviewer_2Abh · 2025-08-05
> >
> > Weakness 1: My comment regarding credit assignment still stands. Without it, the work lacks enough novelty (and improvements)
> >
> > Weakness 2: The variance in performance improvements is still quite high. While the authors claim Llama3-8B-Instruct: ReMA → +20.00% (2.50% → 22.50%), they also have results showing that Llama 3.1 8B instruct drops in performance on the same AMC benchmark. Furthermore, I find it highly surprising that the default GRPO baselines on Qwen 2.5 7B Instruct do not improve performance, and this is not consistent with baselines in other literature such as (https://arxiv.org/pdf/2505.22660). This also applies to the AIME and AMC results, which have very high variance due to low amounts of questions. Other papers in literature with Qwen baselines are https://arxiv.org/pdf/2505.03335.
> >
> > Moreover, I think that before any claims made about meta-thinking and training for the same being important, there should be important baselines and ablations that test other types of agents in the multi-agent setup. Question 2 is not appropriately answered. I believe a simple baseline with a fixed meta-thinking model and RL training another agent conditioned on this input and the question is very important.
> >
> > Unfortunately I will be unable to increase my score and believe that the work still has some issues before it is ready for publication.

---

> ### Author Response · Authors · 2025-08-06
> **Response to Reviewer 2Abh (Part 1/3)**
>
> We thank the reviewer for the constructive feedback and address the additional concerns below.
>
> ### **Weakness 1: Lack of novelty because of no credit assignment**
> Thank you for raising the question about credit assignment. We would like to restate our contributions. We are the first to introduce a multi-agent meta-thinking reasoning process and extend it to a multi-turn interaction setting. To support this framework, we adjust the reinforcement learning algorithm and introduce a turn-level ratio to stabilize training. Our extensive experiments and ablation study demonstrate the effectiveness of both the framework and the training method.
>
> Although we do not invoke classical RL credit‐assignment algorithms in our experiments, our additionally proposed bi‐level framework in Appendix C.5 achieves the same effect. This method is not designed to credit assignment, but we can extract the performance gain of meta-thinking and reasoning agents directly during training, like Monte Carlo state value estimation method in the papers you recommanded.
>
> We will incorporate additional credit-assignment techniques, such as Monte Carlo estimation of state values, as previously proposed, in our future version.
>
> ### **Weakness 2: Improvement is not significant**
>
> We appreciate your detailed inspection on our experiments and insightful feedback. Here we present our detailed response w.r.t. your comments:
>
> > The variance in performance improvements is still quite high. ReMA has performance improvements as well as drop on different benchmarks.
>
> We acknowledge that the performance improvement is not particularly significant. As noted in lines 208–210 of our manuscript, it is **inherently challenging to achieve substantial gains** when starting from a well-optimized instruct model. Given that both ReMA and the baselines are trained on a split of the MATH dataset and evaluated on separate test sets, it is reasonable that ReMA shows improvements on some benchmarks while experiencing drops on others. This is precisely why we introduced a diverse set of test benchmarks and reported the **average performance improvement** across them.

---

> ### Author Response · Authors · 2025-08-06
> **Response to Reviewer 2Abh (Part 2/3)**
>
> > Furthermore, the default GRPO baselines on Qwen 2.5-7B-Instruct do not improve performance, and this is not consistent with baselines in other literature such as [1] and [2].
>
> In fact, we use REINFORCE++ in our experiments in Table 1. Different algorithms may lead to different performance, and we successfully demonstrate the performance of our framework based on the same underlying optimization algorithm. We also attach a comparison with GRPO as follows for your reference, hope this resolves your concern:
>
> **GRPO performance comparison at the final step of 400**
>
> | methods       | amc23 | olympiadbench | gsm8k | MATH500 | aime24 | aime25 | minerva_math | gaokao2023en | Avg  |
> |---------------|-------|----------------|-------|---------|--------|--------|---------------|---------------|--------|
> | **VRP-GRPO**  | 50.00% | 44.89%          | 92.49% | 76.80%   | 10.00%  | 10.00%  | 38.24%         | 62.34%         | **48.09%**  |
> | **ReMA-GRPO** | 57.50% | 44.74%          | 92.19% | 78.20%   | 13.33%  | 13.33%  | 37.50%         | 66.23%         | **50.38%**  |
>
> Due to time constraints, we compared VRP(single-agent GRPO) and ReMA-GRPO (parameter sharing) under a single-turn setting, using the same hyperparameters and trained Qwen2.5-7B-Instruct on the MATH dataset. To illustrate performance trends, we report the sliding average test accuracy at every 100 steps (see the table below for more details).
>
> We observe that single-agent GRPO starts with a higher performance, showing the strength of the base instruct model. However, its performance fluctuates during training and shows limited overall improvement.
>
> In contrast, ReMA-GRPO starts slightly lower but improves steadily and consistently, eventually surpassing GRPO on average. This demonstrates ReMA’s stronger stability and generalization over time, with stable growth and the potential for further improvement after training more steps.
>
> Regarding the referenced literature, for [1], we only observe GRPO training results without verified rewards, which fall outside the scope of our work. Nevertheless, we will include a discussion of this work in our manuscript. As for [2], the results in Table 1 clearly show that **training from an instruct model also yields marginal improvements** (e.g., AceCoderRM: 46.7 → 47.9, CodeR1-LC2k: 46.7 → 48.0).
>
> **GRPO training performance on test datasets per 100 steps**
> **VRP-GRPO (Qwen2.5-7b-instruct)**
> | step | amc23 | olympiadbench | gsm8k | MATH500 | aime24 | aime25 | minerva_math | gaokao2023en | Avg |
> |-----|----|----|----|----|----|----|----|----|----|
> | 0 | 50.00% | 39.70% | 91.28% | 74.60% | 16.67% | 3.33% | 34.56% | 63.90% | 46.75% |
> | 100 | 52.50% | 41.04% | 91.74% | 78.00% | 10.00% | 3.33% | 38.97% | 62.60% | 47.27% |
> | 200 | 50.00% | 43.26% | 92.04% | 77.00% | 6.67% | 6.67% | 38.60% | 62.34% | 47.07% |
> | 300 | 65.00% | 43.41% | 92.65% | 76.80% | 13.33% | 10.00% | 39.34% | 65.45% | 50.75% |
> | 400 | 50.00% | 44.89% | 92.49% | 76.80% | 10.00% | 10.00% | 38.24% | 62.34% | 48.09% |
>
> **ReMA-GRPO (Qwen2.5-7b-instruct)**
> | step | amc23 | olympiadbench | gsm8k | MATH500 | aime24 | aime25 | minerva_math | gaokao2023en | Avg |
> |-----|------|--------------|-------|--------|--------|--------|--------------|--------------|------|
> | 0   | 57.50% | 37.19% | 88.93% | 72.80% | 13.33% | 6.67% | 36.03% | 58.44% | 46.36% |
> | 100 | 55.00% | 39.70% | 91.51% | 76.60% | 13.33% | 6.67% | 36.40% | 62.60% | 47.73% |
> | 200 | 55.00% | 45.48% | 91.81% | 79.00% | 13.33% | 10.00% | 37.50% | 62.34% | 49.31% |
> | 300 | 57.50% | 43.56% | 91.96% | 78.80% | 20.00% | 13.33% | 35.29% | 65.19% | 50.71% |
> | 400 | 57.50% | 44.74% | 92.19% | 78.20% | 13.33% | 13.33% | 37.50% | 66.23% | 50.38% |

---

> ### Author Response · Authors · 2025-08-06
> **Response to Reviewer 2Abh (Part 3/3)**
>
> ### **Question 2: Baselines of other type of agents and setting of fixing meta-thinking agent training reasoning agent**
>
> Thank you for the clarification. First, as requested, we have added Multi-Agent Debate and Self-Refine as additional baselines, and ReMA outperforms both on the majority of tasks. We hope the baselines and performance satisfy your requirements. Second, the baseline that fixes a meta-thinking module and then RL-trains a second agent corresponds exactly to the third setting in Section 4.2.1. In that setting, we first SFT a base model with GPT-4o’s meta-thinking instructions, freeze it, and then RL-train a separate base model for low-level reasoning with the meta-instruction of the fixed meta-thinking agent. Comparing this to the first setting, where we directly RL-train a base model, shows that meta-thinking yields the performance gains illustrated in Figure 3. We apologize for the confusing description and will revise the manuscript accordingly.
>
>
>
>
>
> **Reference**
>
> [1] Prabhudesai, Mihir, et al. "Maximizing Confidence Alone Improves Reasoning." arXiv:2505.22660 (2025).
>
> [2] Zhao, Andrew, et al. "Absolute zero: Reinforced self-play reasoning with zero data." arXiv:2505.03335 (2025).
>
> ---
>
> We hope the above clarifications have addressed your concerns. If there are any remaining questions or points that require further explanation, we would be happy to provide more details.
> If you find that we have adequately addressed your comments, we kindly ask you to consider revisiting your evaluation. We sincerely appreciate your time and constructive feedback.

---

### Note · Authors · 2025-08-15

We sincerely thank the AC and reviewers for constructive discussions. Here is a concise summary of our responses to the main concerns.

**Strengths noted**
Novel separation of meta-thinking from execution with stable multi-agent RL. *(5jA5, uS42, ydZo, 2Abh)*
Consistent gains on challenging benchmarks with comprehensive experiments/ablations. *(ydZo, 5jA5, uS42)*
Effective stability mechanisms (turn-level ratio clipping, parameter sharing). *(2Abh, uS42)*

We appreciate the reviewers’ active engagement in the discussion. Three reviewers confirmed that their concerns were addressed and responded positively. We thank reviewers 5jA5 and uS42 for raising their recommendation to 'Accept'. For the remaining reviewer, we provided detailed two-round replies covering all points, and sincerely hope our clarifications will be considered.

**Additional experiments**
- **Baselines** *(5jA5, ydZo, uS42)*: Added **Multi-Agent Debate** and **Self-Refine**; ReMA attains higher averages across three model families (Qwen-2.5-7B-Inst: **53.09** vs **50.93/50.83**; Llama-3.1-8B-Inst: **36.97** vs **28.61/20.40**; Llama-3-8B-Inst: **26.73** vs **19.55/18.12**).
- **GRPO comparison** *(2Abh)*: we conducted extra experiements on single-turn GRPO and the results show **ReMA-GRPO outperforms single-agent GRPO** on final average and stability under identical budgets.
- **Fixed meta-agent baseline** *(2Abh, ydZo)*: Already covered in Section 4.2.1 (third setting): freeze  an SFTed meta-agent and train reasoner via RL. Fig. 3 shows gains from high-quality meta-instructions + coordinated training.

**Fragility & training stability** *(2Abh, uS42)*
Our turn-level ratio clip mitigates collapse and sustains training, and parameter sharing speeds up training as shown in Fig. 6.

**Significance & variance** *(2Abh)*
We recognize the high variance in small hard sets and therefore report **multi-benchmark averages**, while still observing clear gains on AMC23 (Llama3-8B-Inst: 2.5->22.5) and AIME24 (Qwen2.5-7B-Inst: 6.67->20.00). Despite limited headroom from strong instruct models, our improvements remain **robust** across benchmarks.

**Credit assignment** *(2Abh)*
ReMA’s novelty lies in explicitly decoupling meta-thinking from reasoning execution with stable multi-agent RL. In our experiments, we used a standard sparse reward setting. As future work, we propose three credit-assignment improvements: turn-value Monte Carlo, baseline reuse, and tabular value estimation.

---

### Decision · Program_Chairs · 2025-09-17

**Decision:**

Accept (poster)

**Comment:**

ReMA separates meta-thinking from reasoning using two agents trained together with multi-agent RL. A high-level agent handles strategic oversight while a low-level agent carries out reasoning steps. The two coordinate through RL. The authors show improvements on reasoning-heavy benchmarks like AMC23, AIME24, GSM8K, and MATH500, and provide ablation studies and new baselines like Multi-Agent Debate and Self-Refine to highlight what the meta-thinking layer contributes.

The reviews started off mixed. Some reviewers were positive, calling the separation of meta-thinking from execution novel and appreciating the experiments and ablations. Others were skeptical about fragility, credit assignment, and whether the gains are statistically meaningful or just variance across small test sets. After rebuttal, three reviewers were satisfied, noting the added baselines and clarifications, and raised their scores to accept. One reviewer remained unconvinced, stressing that stronger credit-assignment methods and more careful comparisons are still missing.

The paper asks how to make LLMs explicitly reason about their own reasoning, and makes a concrete step toward answering it. The method is creative, the experiments are fairly comprehensive, and most reviewers ended up leaning positive after discussion. At the same time, the remaining concerns about stability, variance, and ablation depth suggest the work is not yet definitive. To me the contribution feels solid and timely enough for acceptance as a poster.